# PGRF-Net: A Prototype-Guided Relational Fusion Network for Diagnostic Multivariate Time-Series Anomaly Detection

**Jahoon Jeong & Hyunsoo Yoon**[*]
Department of Industrial Engineering, Yonsei University, Seoul, Republic of Korea
{imagenet151, hs.yoon}@yonsei.ac.kr

## Abstract

Multivariate time-series anomaly detection (MTSAD) faces a critical trade-off between detection performance and model transparency. We propose PGRF-Net, a novel architecture designed to achieve competitive performance while providing structured evidence to support diagnostic insights. At its core, PGRF-Net uses a Multi-Faceted Evidence Extractor that combines prototype learning with the discovery of dynamic relational structures between variables. This extractor generates four distinct types of anomaly evidence: predictive deviation, structural changes in learned variable dependencies, contextual deviation from normal-behavior prototypes, and the magnitude of localized spike events. This evidence is then processed by a Gated Evidence Fusion Network, which learns to weigh each source via data-driven gating. PGRF-Net is trained via a two-stage unsupervised strategy for robust extractor learning and subsequent fusion tuning. Extensive experiments on five public MTSAD benchmarks demonstrate its competitive or superior detection performance. Importantly, by decomposing the final anomaly score into these four evidence types, our model facilitates diagnostic analysis, offering a practical step towards more interpretable, evidence-based MTSAD.

## 1 Introduction

Multivariate time-series anomaly detection (MTSAD) is essential in high-stakes domains such as industrial IoT and healthcare, where timely detection prevents failures and losses. Deep models have advanced detection performance (Xu et al., 2022; Wang et al., 2023; Song et al., 2023), yet most remain black boxes: they output a single anomaly score with little insight into its cause, hindering trust and adoption. Moreover, anomalies are heterogeneous—ranging from point spikes to contextual shifts and relational changes—so methods specialized for one type often miss others. This diversity demands a diagnostic framework that not only flags anomalies but also differentiates their underlying nature (Chandola et al., 2009; Zong et al., 2018; Audibert et al., 2020; Blázquez-García et al., 2021; Xu et al., 2022).

To address these limitations, we propose PGRF-Net (Prototype-Guided Relational Fusion Network), a novel architecture that reframes anomaly detection as a diagnostic process based on multi-faceted evidence aggregation. The key idea of PGRF-Net lies not in any single component, but in the synergistic integration of a Multi-Faceted Evidence Extractor and a Gated Evidence Fusion Network into a unified diagnostic framework. The extractor is designed to align with the practical needs of system operators by evaluating four complementary evidence sources: predictive deviation, structural changes in learned variable dependencies, contextual deviation from normal-behavior prototypes, and localized spike events. The fusion network then integrates this evidence, learning a data-driven strategy to dynamically weigh each source. These sources provide complementary views; we do not assume time invariance of the low-frequency component. The entire system is optimized via a two-stage unsupervised strategy, ensuring reliable, data-driven anomaly detection while reducing reliance on manual heuristics.

The main contributions of this paper are:

---

[*]Corresponding author.

- **A Diagnostic Framework for Anomaly Attribution.** We propose PGRF-Net, an architecture that moves beyond single-score detection by decomposing anomalies into four evidence types: predictive, structural, contextual, and spike. This decomposition is directly motivated by the practical need for diagnostic analysis, offering decomposed anomaly attribution, unlike traditional black-box models. To our knowledge, this is one of the early attempts to reframe MTSAD as a diagnostic process rather than a mere detection task, by decomposing anomalies into multiple evidence types within a unified framework.

- **Unsupervised Gated Fusion Mechanism.** We introduce a Gated Evidence Fusion Network that learns to optimally weigh the decomposed evidence in a data-driven, unsupervised manner. This eliminates heuristic-based tuning and allows for robust detection across diverse anomaly types.

- **Competitive Performance with Interpretability.** PGRF-Net achieves competitive or superior detection performance compared to state-of-the-art methods across five widely-used public benchmarks, while uniquely providing actionable, decomposed explanations for anomaly attribution.

## 2 RELATED WORK

### 2.1 MTSAD APPROACHES

Modern MTSAD approaches are diverse. **Reconstruction-based** methods employ Autoencoders (AEs) (Audibert et al., 2020), Variational Autoencoders (VAEs) (Su et al., 2019), and GANs (Li et al., 2019) to flag instances with high reconstruction error. **Prediction-based** methods rely on temporal models such as LSTMs (Malhotra et al., 2015) and Transformers (Xu et al., 2022; Wu et al., 2023) to detect large forecast errors. More recent paradigms include **contrastive learning**, which separates normal and abnormal patterns (Yang et al., 2023; Tuli et al., 2022), and **diffusion models**, which employ denoising processes for reconstruction or imputation (Xiao et al., 2023; Chen et al., 2023; Pintilie et al., 2023). Despite strong performance, most of these methods produce a single anomaly score, limiting their utility for diagnostic analysis.

### 2.2 PROTOTYPE LEARNING FOR INTERPRETABILITY

Prototype learning improves interpretability by capturing representative normal patterns, often stored in memory banks (Gong et al., 2019). Anomalies are then identified by their deviation from these prototypes. This idea has evolved into sophisticated memory modules, as in MNAD (Park et al., 2020) and MEMTO (Song et al., 2023), which enhance reliability by avoiding the reconstruction of abnormal inputs (Shen et al., 2025). While effective for modeling contextual normality, PGRF-Net extends this paradigm by introducing prototype-guided channels not only for contextual but also for structural deviations, thereby offering a broader diagnostic perspective.

### 2.3 DEPENDENCY-AWARE ANOMALY DETECTION

Anomalies in multivariate time series often manifest through violations of inter-variable dependencies. Early statistical approaches relied on Granger causality tests (Qiu et al., 2012), while recent deep models have advanced dependency modeling in three complementary directions: association-based discrepancies (e.g., Anomaly Transformer (Xu et al., 2022)), causality-driven discovery (e.g., GCAD (Liu et al., 2025), CAROTS (Kim et al., 2025)), and dynamic graph reasoning (e.g., GSC-MAD (Zhang et al., 2024), Causalformer (Kong et al., 2024)). These methods highlight the importance of modeling variable dependencies but often treat them as the *primary* anomaly signal. In contrast, PGRF-Net incorporates dependency evidence as one of four complementary sources, enabling a more balanced and diagnostic view of anomalies.

## 3 METHODOLOGY

This section details PGRF-Net, a novel architecture designed to enhance both performance and diagnostic transparency in MTSAD. Illustrated in Figure 1, PGRF-Net is composed of two core components trained in a two-stage unsupervised process: a **Multi-Faceted Evidence Extractor**

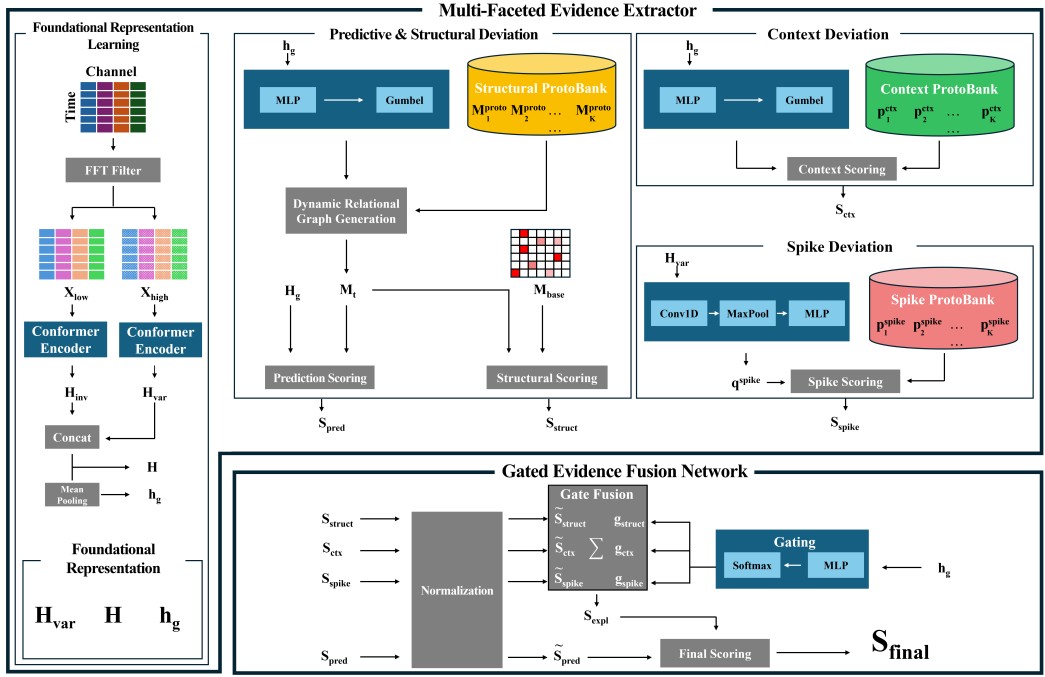

Figure 1: The overall architecture of PGRF-Net. The model learns a foundational representation via frequency decomposition and Conformer encoders, then generates four diagnostic evidence scores (*predictive*, *structural*, *contextual*, and *spike* deviations). A Gated Evidence Fusion Network learns to combine these scores to produce the final anomaly score.

that decomposes anomalies into distinct causes, and a **Gated Evidence Fusion Network** that learns to combine them for a final decision.

## 3.1 STAGE 1: MULTI-FACETED EVIDENCE EXTRACTOR

The extractor, trained in the first stage, serves as the foundational engine of PGRF-Net. It is designed not only for robust representation learning but also for generating four distinct diagnostic sources of anomaly evidence.

### 3.1.1 FOUNDATIONAL REPRESENTATION LEARNING

To effectively model multi-scale dynamics, we first decompose an input window $\mathbf{X} \in \mathbb{R}^{w \times N}$, where $w$ is the window size and $N$ is the number of variables, into its low-frequency (trend-like) and high-frequency (transient) components, $\mathbf{X}_{low}$ and $\mathbf{X}_{high}$, via an FFT-based filter.

This split provides two complementary views and does not assume time invariance of the low-frequency component. Each component is processed by a dedicated Conformer encoder (Gulati et al., 2020), which adeptly captures both global and local patterns:

$$\mathbf{H}_{inv} = \text{Conformer}(\mathbf{X}_{low}), \quad \mathbf{H}_{var} = \text{Conformer}(\mathbf{X}_{high}). \tag{1}$$

The two representations are concatenated and linearly projected to yield a fused feature map:

$$\mathbf{H} = \text{Linear}\big([\mathbf{H}_{inv}; \mathbf{H}_{var}]\big), \tag{2}$$

from which we derive a global context vector via temporal pooling:

$$\mathbf{h}_g = \text{MeanPool}(\mathbf{H}). \tag{3}$$

Here, $\mathbf{H}$ serves as the general fused feature map, while $\mathbf{H}_{var}$ is also retained as a spike-sensitive representation for transient evidence extraction. Together with $\mathbf{h}_g$, they form the cornerstone for subsequent evidence generation.

### 3.1.2 LEARNABLE PROTOTYPE BANKS FOR NORMALCY MODELING

Instead of learning a single, monolithic model of normal behavior, PGRF-Net maintains three distinct, learnable prototype banks, each storing a vocabulary of representative exemplars corresponding to different facets of normalcy. Here, $K$, $K_c$, and $K_s$ denote the number of prototypes in the structural, context, and spike banks, respectively.

**Structural Prototype Bank** ($\mathcal{B}_{struct} \in \mathbb{R}^{K \times N \times N}$). Each slice $\mathbf{M}_k^{proto} \in \mathbb{R}^{N \times N}$ encodes a canonical dependency pattern between the $N$ variables, capturing inter-variable *structural relations*.

**Context Prototype Bank** ($\mathcal{B}_{ctx} \in \mathbb{R}^{K_c \times d_{\text{model}}}$). Each vector $\mathbf{p}_k^{ctx}$ represents a distinct operational state, capturing diverse *modes of normal system behavior*.

**Spike Prototype Bank** ($\mathcal{B}_{spike} \in \mathbb{R}^{K_s \times d_{\text{model}}}$). Each vector $\mathbf{p}_k^{spike}$ encodes an archetypal transient fluctuation, characterizing short-lived but benign *events*.

These banks are initialized randomly and progressively specialized through end-to-end optimization during Stage 1 training.

### 3.1.3 EVIDENCE GENERATION VIA PROTOTYPE COMPARISON

With the learned representations and prototype banks, the model generates four evidence scores. This process begins with generating a dynamic relational graph from $\mathbf{h}_g$, which is used for both prediction and structural deviation analysis.

**Dynamic Relational Graph Generation.** A selector network processes $\mathbf{h}_g$ to determine the most relevant relational structure for the current time window. It computes dynamic weights via an MLP and the Gumbel-Softmax function $\mathcal{G}(\cdot)$, which are then used to create a weighted combination of the prototypes in the structural prototype bank. We view $\mathbf{M}_t$ as a dependency mask. This yields a dynamic relational mask:

$$\mathbf{M}_t = \sum_{k=1}^{K} \mathcal{G}(\text{MLP}_{sel}(\mathbf{h}_g))_k \cdot \mathbf{M}_k^{proto}. \tag{4}$$

We then derive four complementary evidence scores:

- **Predictive Deviation** ($S_{pred}$): From the fused feature map $\mathbf{H} \in \mathbb{R}^{w \times d_{\text{model}}}$, we extract per-variable representations $\mathbf{h}_j$ ($j = 1, \ldots, N$). The parent feature of variable $i$ is computed as:

$$\mathbf{h}_{pa(i)} = \sum_{j=1}^{N} (\mathbf{M}_t)_{ij} \cdot \mathbf{h}_j, \tag{5}$$

  and $\hat{y}_i$ is predicted at the last step using a lightweight linear head on $[\mathbf{h}_i, \mathbf{h}_{pa(i)}]$. The predictive deviation is:

$$S_{pred} = \frac{1}{N} \sum_{i=1}^{N} (y_i - \hat{y}_i)^2. \tag{6}$$

- **Structural Deviation** ($S_{struct}$): Measures how much $\mathbf{M}_t$ deviates from a baseline $\mathbf{M}_{base}$, computed by averaging early-training masks on the normal training split, with the diagonal set to zero and kept fixed for inference. Each prototype $\mathbf{M}_k^{proto}$ is constrained by an acyclicity penalty (Zheng et al., 2018). The deviation is:

$$S_{struct} = \|\mathbf{M}_t - \mathbf{M}_{base}\|_F. \tag{7}$$

- **Contextual Deviation** ($S_{ctx}$): The global context $\mathbf{h}_g$ is compared against the context prototype bank $\{\mathbf{p}_k^{ctx}\}$ to form a weighted normal reference:

$$\mathbf{p}_w^{ctx} = \sum_{k=1}^{K_c} \mathcal{G}(\text{MLP}_{ctx}(\mathbf{h}_g))_k \cdot \mathbf{p}_k^{ctx}. \tag{8}$$

  The deviation is:

$$S_{ctx} = 1 - \frac{\mathbf{h}_g \cdot \mathbf{p}_w^{ctx}}{\|\mathbf{h}_g\|_2 \cdot \|\mathbf{p}_w^{ctx}\|_2}. \tag{9}$$

- **Spike Deviation** ($S_{spike}$): A spike feature $\mathbf{q}_{spike}$ is extracted from the high-frequency map $\mathbf{H}_{var}$:

$$\mathbf{q}_{spike} = \text{MLP}_{spike}(\text{MaxPool}(\text{Conv1D}(\mathbf{H}_{var}))). \tag{10}$$

Its deviation is the minimum distance to the spike prototypes:

$$S_{spike} = \min_k \|\mathbf{q}_{spike} - \mathbf{p}_k^{spike}\|_2. \tag{11}$$

### 3.1.4 Stage 1 Training Objective

The extractor is trained on normal data using a composite loss function designed to build a comprehensive model of normalcy:

$$\mathcal{L}_{\text{Stage1}} = \lambda_{pred}\mathcal{L}_{pred} + \lambda_{struct}\mathcal{R}_{struct} + \lambda_{proto}\mathcal{R}_{proto}. \tag{12}$$

**Prediction Loss ($\mathcal{L}_{pred}$).** We adopt a Focal Loss-style weighting (Lin et al., 2017) on the prediction MSE. For a sample with MSE $m$, define its correctness probability as

$$p_t = 1 - \text{scaled\_MSE}(m), \tag{13}$$

and the loss as

$$\mathcal{L}_{pred} = \alpha_t(1 - p_t)^\gamma \, m, \tag{14}$$

which emphasizes higher MSE samples that better delineate the boundaries of normal behavior.

**Structural Regularization ($\mathcal{R}_{struct}$).** This term ensures relational masks are meaningful and stable:

$$\mathcal{R}_{struct} = \lambda_{l1}\mathcal{L}_{l1} + \lambda_{sparse}\mathcal{L}_{sparse} + \lambda_{acyc}\mathcal{L}_{acyc} + \lambda_{diff}\mathcal{L}_{diff}. \tag{15}$$

- $\mathcal{L}_{l1}$: sparsity on the batch-averaged relational mask.
- $\mathcal{L}_{sparse}$: sparsity directly on prototype masks $\{\mathbf{M}^{(proto,k)}\}$, encouraging compact structural blocks.
- $\mathcal{L}_{acyc}$: DAG constraint (Zheng et al., 2018) on each prototype mask, ensuring that $\mathbf{M}_t$ (a convex combination) inherits acyclicity.
- $\mathcal{L}_{diff}$: temporal stability by penalizing deviations from the baseline mask $\mathbf{M}_{base}$.

**Prototype Regularization ($\mathcal{R}_{proto}$).** A small-weighted clustering loss encourages compact, diverse prototypes:

$$\mathcal{R}_{proto} = \lambda_{ctx1} \mathbb{E}[S_{ctx}] + \lambda_{spike1} \mathbb{E}[S_{spike}]. \tag{16}$$

## 3.2 Stage 2: Gated Evidence Fusion Network

In the second stage, we freeze the feature extractor (Conformer encoders) to stabilize the learned representations, while allowing the prototype banks and the gating module to remain trainable. This ensures that Stage 2 focuses on calibrating evidence attribution without altering the underlying feature space. The fusion network is then trained in an unsupervised manner to activate explanatory channels only when strong evidence arises. By combining pseudo-normal suppression and entropy sharpening, the gating module provides a *data-driven weighting* of evidence sources, replacing heuristic fusion with an instance-dependent mechanism.

### 3.2.1 Gated Fusion Mechanism

A lightweight gating controller (MLP) maps the global context vector $\mathbf{h}_g$ into a softmax weight vector over the explanatory channels, $\mathbf{g} = [g_{struct}, g_{ctx}, g_{spike}]$ with $\mathbf{g} = \text{softmax}(\text{MLP}(\mathbf{h}_g))$. Each deviation score is first normalized to $[0, 1]$ by min–max scaling. The explanatory score is obtained by fusing the structural, contextual, and spike channels:

$$S_{expl} = \sum_{i \in \{struct,ctx,spike\}} \frac{g_i}{\sum_{j \in \{struct,ctx,spike\}} g_j} \cdot \tilde{S}_i \tag{17}$$

The final anomaly score then balances predictive and explanatory evidence via a hyperparameter $\alpha$:

$$S_{\text{final}} = (1 - \alpha)\tilde{S}_{pred} + \alpha S_{expl} \tag{18}$$

Here, $\alpha \in [0, 1]$ serves as a simple trade-off knob: smaller values emphasize prediction error, while larger values highlight explanatory diagnostics.

### 3.2.2 STAGE 2 TRAINING OBJECTIVE

The fusion module is trained without anomaly labels, guided by Stage 1 signals. Its objective is

$$\mathcal{L}_{\text{Stage2}} = \lambda_{ctx2} \, \mathbb{E}[S_{ctx}] + \lambda_{spike2} \, \mathbb{E}[S_{spike}] + \lambda_{sup}\mathcal{L}_{sup} + \lambda_{ent}\mathcal{L}_{ent}. \qquad (19)$$

- **Prototype Tuning.** Refinement losses on $S_{ctx}$ and $S_{spike}$ sharpen the separation of normal clusters, reinforcing diverse yet compact prototypes.
- **Gate Suppression Loss ($\mathcal{L}_{sup}$).** We define the total explanatory gate activation as:

$$G_{\text{expl}} = \sum_{i \in \{struct, ctx, spike\}} g_i \qquad (20)$$

 For pseudo-normal samples $\mathcal{X}_{pn}$ (low prediction error in Stage 1), this loss penalizes $G_{\text{expl}}$:

$$\mathcal{L}_{sup} = \mathbb{E}_{\mathbf{X} \in \mathcal{X}_{pn}}[G_{\text{expl}}] \qquad (21)$$

 thereby encouraging the network to keep gates inactive when no explanation is needed.
- **Entropy Loss ($\mathcal{L}_{ent}$).** Encourages sharper gate activations:

$$\mathcal{L}_{ent} = \mathbb{E}\Big[ -\sum_i g_i \log g_i \Big], \qquad (22)$$

 promoting selective attribution to the most relevant channel.

## 4 EXPERIMENTS

We conduct extensive experiments to validate PGRF-Net's performance and interpretability. Our evaluation addresses two research questions: **(RQ1)** How does PGRF-Net compare with SOTA methods on public benchmarks? **(RQ2)** How do the proposed components contribute to accuracy and diagnostic transparency? We next describe datasets and metrics; then we present benchmark comparisons, ablations, and interpretability analyses.

### 4.1 EXPERIMENTAL SETUP

We evaluate PGRF-Net on five public benchmarks: MSL and SMAP (Hundman et al., 2018), PSM (Abdulaal et al., 2021), SMD (Su et al., 2019), and SWaT (Mathur & Tippenhauer, 2016).

Following established practice (Su et al., 2019; Shen et al., 2025), we report point-adjusted Precision, Recall, and F1, together with threshold-free AUC-ROC and AUC-PR. To provide an additional *event-level* evaluation, we also include Range-Precision (R-P), Range-Recall (R-R), and Range-F1 (R-F1) (Tatbul et al., 2018) in our main result tables. For completeness, we additionally report point-wise (non-adjusted) Precision/Recall/F1 in Appendix C, allowing direct comparison with prior methods that rely on point-wise metrics. All results are averaged across five random seeds. Detailed dataset statistics and evaluation procedures are provided in Appendix A–B.

### 4.2 BASELINES

We compare PGRF-Net with both established models (Deep-SVDD (Ruff et al., 2018), LSTM-VAE (Park et al., 2018), THOC (Shen et al., 2020), OmniAnomaly (Su et al., 2019), InterFusion (Li et al., 2021)) and recent SOTA models (D3R (Wang et al., 2023), DMamba (Chen et al., 2024), DCdetector (Yang et al., 2023), MEMTO (Song et al., 2023), GSC-MAD (Zhang et al., 2024), H-PAD (Shen et al., 2025)).

### 4.3 IMPLEMENTATION DETAILS

PGRF-Net is implemented in PyTorch and trained with Adam (lr=$10^{-4}$). We use a sliding window of $w = 60$ and model dimension $d_{model} = 128$. The training data is further split into train/validation sets with an 8:2 ratio to enable hyperparameter tuning and early stopping. Other hyperparameters follow standard practice and are listed in Appendix D. Thresholds for thresholded metrics are selected on the validation set, and range-based metrics (R-P/R-R/R-F1) directly follow the official implementation of (Tatbul et al., 2018). All experiments are conducted on a single NVIDIA A100 GPU (40GB).

Table 1: Performance comparison on five real-world datasets in terms of Precision (P), Recall (R), and F1-score. All scores are in %. **Best** and second best results are highlighted.

| Model | MSL | | | SMAP | | | PSM | | | SMD | | | SWaT | | | Avg F1 |
|---|---|---|---|---|---|---|---|---|---|---|---|---|---|---|---|---|
| | P | R | F1 | P | R | F1 | P | R | F1 | P | R | F1 | P | R | F1 | |
| Deep-SVDD | 91.92 | 76.63 | 83.58 | 89.93 | 56.02 | 69.04 | 95.41 | 86.49 | 90.73 | 78.54 | 79.67 | 79.10 | 80.42 | 84.45 | 82.39 | 80.97 |
| THOC | 88.45 | 90.97 | 89.69 | 92.06 | 89.34 | 90.68 | 88.14 | 90.99 | 89.54 | 79.76 | 90.95 | 84.99 | 83.94 | 86.36 | 85.13 | 88.01 |
| LSTM-VAE | 85.49 | 79.94 | 82.62 | 92.20 | 67.75 | 78.10 | 73.62 | 89.92 | 80.96 | 75.76 | 90.08 | 82.30 | 76.00 | 89.50 | 82.20 | 81.24 |
| OmniAnomaly | 89.02 | 86.37 | 87.67 | 92.49 | 81.99 | 86.92 | 88.39 | 74.46 | 80.83 | 83.68 | 86.82 | 85.22 | 81.42 | 84.30 | 82.83 | 84.69 |
| InterFusion | 81.28 | 92.70 | 86.62 | 89.77 | 88.52 | 89.14 | 83.61 | 83.45 | 83.52 | 87.02 | 85.43 | 86.22 | 80.59 | 85.58 | 83.01 | 85.70 |
| DCdetector | 92.09 | **98.89** | 95.37 | 94.42 | 98.95 | 96.63 | 97.24 | 97.72 | 97.48 | 86.08 | 85.60 | 85.84 | 93.29 | **100.00** | 96.53 | 94.37 |
| D3R | 91.77 | 94.33 | 93.03 | 92.23 | 96.11 | 94.21 | 93.84 | 99.11 | 96.45 | 87.74 | 96.09 | 91.91 | 83.09 | 83.00 | 83.04 | 91.73 |
| MEMTO | 91.95 | 97.23 | 94.56 | 93.66 | **99.73** | 96.60 | 97.47 | 98.60 | 98.03 | 88.24 | 96.16 | 92.03 | 94.28 | 91.72 | 93.73 | 94.99 |
| DMamba | 93.69 | 64.06 | 76.09 | 95.10 | 52.98 | 68.05 | 98.66 | 82.59 | 89.91 | 92.57 | 54.04 | 68.24 | 94.11 | 86.75 | 90.28 | 78.51 |
| GSC-MAD | 94.19 | 93.09 | 93.63 | 89.57 | 98.35 | 93.76 | 97.97 | 99.14 | 98.89 | 92.25 | 94.42 | 93.32 | **96.73** | 95.11 | 95.91 | 95.10 |
| H-PAD | 94.05 | 96.88 | 95.45 | **96.00** | 98.45 | **97.21** | **98.82** | **99.41** | **99.12** | 92.86 | 98.20 | 95.45 | 94.34 | **100.00** | 97.09 | 96.86 |
| PGRF-Net (Ours) | **96.55** | 98.50 | **97.53** | 93.10 | 98.15 | 95.55 | 98.40 | 99.10 | 98.85 | **96.50** | **99.80** | **97.75** | 96.10 | 98.25 | **97.37** | **97.41** |

Table 2: Performance comparison on five real-world datasets in terms of AUC-ROC and AUC-PR. All scores are in %. **Best** and second best results are highlighted.

| Model | MSL | | SMAP | | PSM | | SMD | | SWaT | | Avg | |
|---|---|---|---|---|---|---|---|---|---|---|---|---|
| | AUC-ROC | AUC-PR | AUC-ROC | AUC-PR | AUC-ROC | AUC-PR | AUC-ROC | AUC-PR | AUC-ROC | AUC-PR | AUC-ROC | AUC-PR |
| LSTM-VAE | 52.12 | 4.52 | 50.83 | 4.19 | 49.15 | 40.22 | 50.05 | 4.15 | 49.59 | 4.13 | 50.35 | 11.44 |
| D3R | **65.26** | **16.99** | 41.35 | 10.62 | 50.03 | 26.31 | 64.20 | 12.24 | 56.65 | 13.30 | 55.50 | 15.89 |
| DCdetector | 50.06 | 10.61 | 48.87 | 12.48 | 49.83 | 27.64 | 48.77 | **41.16** | 49.74 | 11.60 | 49.45 | 20.70 |
| H-PAD | 59.99 | 15.06 | 59.13 | 15.30 | 75.01 | 51.83 | **76.49** | 14.05 | 81.54 | 53.99 | 70.43 | 30.05 |
| PGRF-Net (Ours) | 64.50 | 16.80 | **60.10** | **16.55** | 76.25 | 52.10 | 75.50 | 22.50 | 83.10 | 54.50 | 71.89 | 32.49 |

## 4.4 OVERALL PERFORMANCE COMPARISON (RQ1)

Tables 1 and 2 compare PGRF-Net with a broad range of established and state-of-the-art baselines across five public datasets.

**F1-Score Analysis.** As shown in Table 1, PGRF-Net achieves the highest average F1-score (97.41%) among all methods, ranking first on three out of five datasets (MSL, SMD, SWaT) and second on the others. This improvement highlights the benefit of integrating multi-faceted evidence rather than relying on a single anomaly perspective. Notably, our model balances precision and recall, avoiding the high-recall but low-precision tradeoff observed in DCdetector and MEMTO. All F1-scores in this subsection refer to the standard point-adjusted F1 metric; event-level Range-F1 is analyzed separately in Table 3.

**AUC Score Analysis.** The threshold-independent metrics in Table 2 further confirm the robustness of PGRF-Net. It achieves the best average AUC-PR (32.49%) and a strong AUC-ROC, crucial in imbalanced anomaly detection. Compared to the prototype-based model H-PAD, PGRF-Net provides higher recall while preserving interpretability via structured fusion.

**Range-based Evaluation.** Table 3 reports event-level Range-Precision (R-P), Range-Recall (R-R), and Range-F1 (R-F1), which evaluate anomaly segments as coherent events rather than isolated points. Across the five datasets, PGRF-Net consistently ranks at or near the top: it achieves the best Range-F1 on MSL, SMD, and SWaT, achieves the second-best score on SMAP, and remains competitive on PSM.

Taken together, these results demonstrate that PGRF-Net not only achieves state-of-the-art detection accuracy but also provides consistent reliability across datasets with very different anomaly characteristics.

Table 3: Event-level range-based metrics (R-P, R-R, R-F1). **Best** and second best are highlighted.

| Model | MSL | | | SMAP | | | PSM | | | SMD | | | SWaT | | | Avg R-F1 |
|---|---|---|---|---|---|---|---|---|---|---|---|---|---|---|---|---|
| | R-P | R-R | R-F1 | R-P | R-R | R-F1 | R-P | R-R | R-F1 | R-P | R-R | R-F1 | R-P | R-R | R-F1 | |
| THOC | 85.40 | 82.33 | 83.83 | 88.12 | 79.24 | 83.46 | 84.10 | 71.52 | 77.24 | 82.30 | 86.14 | 84.18 | 77.92 | 81.30 | 79.57 | 81.66 |
| OmniAnomaly | 78.91 | 93.42 | 85.46 | 87.10 | 86.25 | 86.66 | 80.24 | 82.10 | 81.16 | 87.22 | 86.91 | 87.06 | 79.10 | 86.22 | 82.48 | 84.56 |
| InterFusion | 89.15 | **97.82** | 93.25 | 92.72 | 97.94 | **95.25** | 95.82 | 96.74 | 96.27 | 86.40 | 85.11 | 85.75 | 90.52 | **98.31** | 94.23 | 92.95 |
| DCdetector | 88.44 | 92.31 | 90.33 | 90.25 | 93.44 | 91.82 | 92.10 | 97.22 | 94.59 | 88.92 | 94.84 | 91.75 | 81.04 | 81.11 | 81.07 | 89.91 |
| D3R | 89.72 | 95.44 | 92.48 | 91.84 | **98.10** | 94.85 | 96.52 | 97.35 | 96.93 | 90.52 | 97.10 | 93.69 | 92.22 | 90.45 | 91.33 | 93.86 |
| MEMTO | 92.40 | 91.91 | 92.15 | 88.60 | 96.70 | 92.43 | **97.20** | **98.40** | **97.79** | 93.44 | 97.01 | 95.46 | **95.10** | 94.22 | 94.66 | 94.50 |
| **PGRF-Net (Ours)** | **95.85** | 96.92 | **96.38** | **93.02** | 97.42 | 95.17 | 95.10 | 96.92 | 96.00 | **95.92** | **98.44** | **97.16** | 94.22 | 97.20 | **95.68** | **96.08** |

Table 4: Ablation study on core architectural modules. "w/ FD" denotes with Frequency Decomposition (at different low-frequency ratios). All metrics are Avg. F1-Score (%).

| Encoder | w/o FD | w/ FD (Low Freq. Ratio) | | |
|---|---|---|---|---|
| | | 10% | 20% | 30% |
| MLP-Mixer | 92.88 | 93.81 | 94.15 | 93.97 |
| Transformer | 94.71 | 95.80 | 96.05 | 95.91 |
| PatchTST | 95.34 | 96.52 | 96.82 | 96.75 |
| **Conformer** | 95.12 | 97.15 | **97.41** | 97.35 |

Table 5: Impact of two-stage training and gated fusion. Static Fusion ratios denote weights for $(S_{pred} : S_{struct} : S_{ctx} : S_{spike})$.

| Stage | Model Variant | Avg F1 |
|---|---|---|
| 1 | $S_{pred}$ only | 92.51 |
| | Static Fusion (1:0:1:1) | 94.13 |
| | Static Fusion (1:1:0:1) | 94.58 |
| | Static Fusion (1:1:1:0) | 95.34 |
| | Static Fusion (1:1:1:1) | 96.82 |
| | Max Pooling (of 4 scores) | 92.32 |
| 1+2 | **PGRF-Net** | **97.41** |

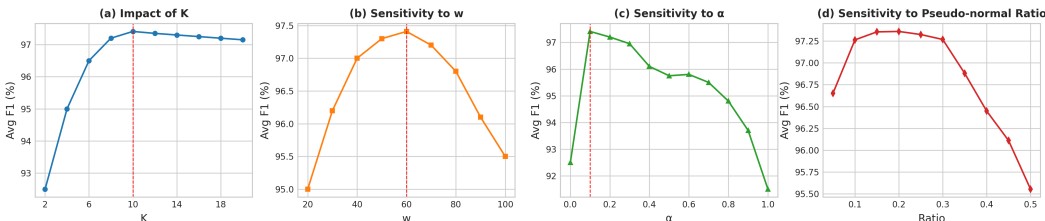

Figure 2: In-depth analysis of key components. (a) Impact of the number of prototypes K. (b) Sensitivity to window size $w$. (c) Sensitivity to fusion weight $\alpha$. (d) Sensitivity to the pseudo-normal ratio for Stage-2 training.

## 4.5 IN-DEPTH ANALYSIS AND ABLATION STUDY (RQ2)

We further analyze the contributions of PGRF-Net's core components through ablation studies.

### 4.5.1 EFFECTIVENESS OF CORE ARCHITECTURAL DESIGN.

We validate our architectural choices—the dual-stream frequency decomposition (FD) and the Conformer encoder—by replacing them with strong alternatives (MLP-Mixer, Transformer, PatchTST), both with and without FD. As shown in Table 4, applying FD consistently improves performance across all backbones, confirming that separating slow and fast dynamics provides complementary features. Among all encoders, the Conformer achieves the best performance (97.41% F1), highlighting its ability to capture both local and global temporal dependencies, which synergize well with FD by aligning temporal receptive fields with frequency-separated signals.

### 4.5.2 IMPACT OF TWO-STAGE TRAINING AND GATED FUSION.

To assess the benefit of the two-stage strategy, we compare the full PGRF-Net with Stage-1-only variants under different static fusion ratios. Results in Table 5 show that while static fusion already outperforms the prediction-only baseline, its performance is highly sensitive to fixed weights and degrades when evidence channels are imbalanced or removed. By contrast, the Stage-2 gated fusion consistently yields the best performance (97.41%), demonstrating that learning input-dependent weights is essential for stability beyond fixed heuristics.

### 4.5.3 HYPERPARAMETER SENSITIVITY.

Figure 2 presents sensitivity analyses for key hyperparameters. Performance is stable across a broad range, peaking at $K = 10$ prototypes, window size $w = 60$, and fusion weight $\alpha = 0.1$. The pseudo-normal ratio is effective between 0.1 and 0.3. Variance across five seeds is very small, confirming robustness. Note that sensitivity plots aggregate results across datasets, while ablation tables (Table 4, Table 5) report single-dataset studies, explaining minor discrepancies in absolute F1-scores and preventing misinterpretation as methodological inconsistency.

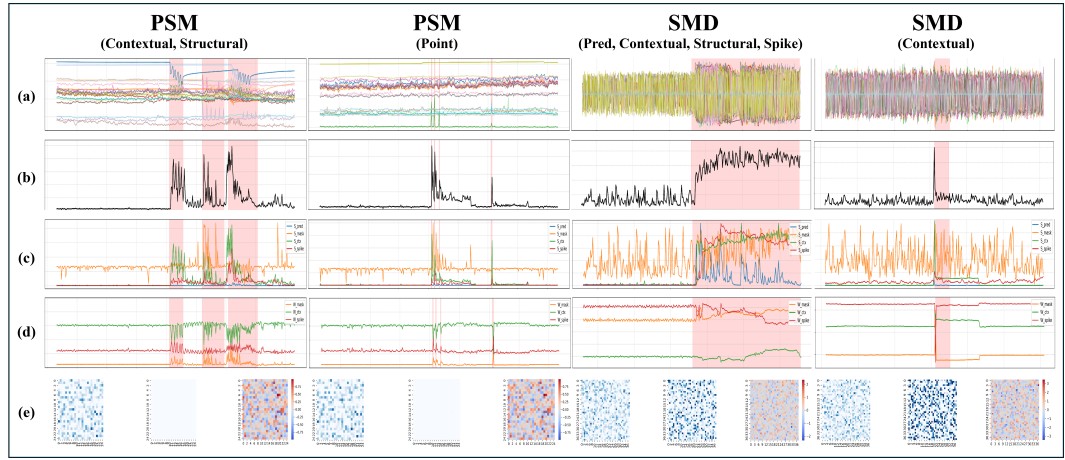

Figure 3: Case studies of PGRF-Net's interpretability on four anomaly events from the benchmark datasets. Each column presents a complete diagnostic report with five aligned views (see Section 4.6): (a) raw signals with time on the x-axis and sensor values on the y-axis, (b) final anomaly score, (c) decomposed evidence attribution ($S_{pred}$, $S_{ctx}$, $S_{spike}$, $S_{struct}$), (d) adaptive fusion weights, and (e) relational-level changes (red = strengthened, blue = weakened). Symbols in Row (e) mark altered dependencies, directly linking anomaly events to structural shifts. Together, these views demonstrate how PGRF-Net translates detection into actionable reasoning. **Note: $S_{mask}$ in plots corresponds to Structural Deviation ($S_{struct}$).**

## 4.6 INTERPRETABILITY ANALYSIS

A central goal of PGRF-Net is to move beyond black-box detection and serve as a transparent diagnostic assistant. Rather than outputting only an anomaly score, our framework produces a structured five-panel report designed to explain both *what* is anomalous and *why*:

- **Raw Signal (Row a).** Input time series with anomaly regions highlighted for localization.
- **Final Score (Row b).** Aggregated anomaly score curve that confirms severity and boundaries.
- **Decomposed Evidence (Row c).** Attribution to predictive, structural, contextual, and spike evidence, revealing the anomaly's type.
- **Adaptive Weights (Row d).** Dynamic fusion coefficients showing the model's internal reasoning and emphasis shifts across evidence sources.
- **Relational-Level Insight (Row e).** Fine-grained diagnostic analysis by contrasting anomalous relational masks ($M_t$) with baseline masks ($M_{base}$).

Representative case studies are shown in Figure 3, and extended examples across benchmarks are in Appendix E.1.

While real-world benchmarks demonstrate that PGRF-Net produces semantically meaningful decomposed evidence, these datasets lack ground-truth anomaly *types* (e.g., spike vs. contextual vs. structural). Therefore, to quantitatively validate that each evidence channel ($S_{struct}$, $S_{ctx}$, $S_{spike}$) responds selectively to the intended anomaly mechanism, we perform controlled synthetic experiments, as detailed in Appendix E.3.

## 4.7 COMPLEXITY AND EFFICIENCY

To assess the practical deployability of PGRF-Net, we additionally evaluate its computational complexity and runtime efficiency under a standardized setup. For a fair comparison, all models were benchmarked on the PSM dataset (25 channels, 132,481 training time steps and 87,841 test time steps) using a single NVIDIA A100 GPU. As summarized in Table 6, our proposed PGRF-Net demonstrates parameter efficiency with only 2.25M parameters, which is significantly fewer than prior models like MEMTO (5.39M). The model requires just 0.005 GPU-hours for training and

achieves an inference latency of 0.059 ms per step, supporting real-time deployment scenarios. This demonstrates that our framework achieves strong performance with high computational efficiency.

Table 6: Model complexity and efficiency, benchmarked on a single NVIDIA A100 GPU.

| Model | #Params (M) | Train Time (GPU-h) | Inference (ms/step) |
|---|---|---|---|
| D3R | 3.152 | 0.047 | 0.141 |
| MEMTO | 5.39 | **0.004** | **0.043** |
| PGRF-Net (Ours) | **2.25** | 0.005 | 0.059 |

## 5 LIMITATIONS

While PGRF-Net substantially improves diagnostic interpretability for MTSAD, several limitations remain that highlight directions for future work.

**Human-centered evaluation.** Although our quantitative tests (selectivity, synthetic isolation, stability) demonstrate semantic alignment of evidence channels, we do not include human-centered or practitioner-based evaluations. Assessing the real-world usefulness for operators remains future work.

**Training contamination.** Real-world training data may contain unlabeled anomalies, and our method does not explicitly model heavy contamination. Extending prototype formation and gating to be contamination-aware is a promising future direction.

**DAG constraint and feedback loops.** Our structural evidence relies on DAG regularization, which improves stability and interpretability but cannot capture feedback loops that frequently arise in cyber-physical systems. Extending structural modeling to cyclic or feedback-aware graphs remains an open direction.

**Baseline mask sensitivity.** The baseline structural mask $M_{\text{base}}$ is derived from early training under a stationary-normal assumption. In environments with multi-regime or drifting normal dynamics, a single averaged baseline may be insufficient; regime-conditioned or multi-prototype baselines would improve flexibility.

## 6 CONCLUSION

We introduced PGRF-Net, a prototype-guided relational fusion network that reframes multivariate time-series anomaly detection as a diagnostic task rather than mere flagging. Its design combines a Multi-Faceted Evidence Extractor—decomposing anomalies into predictive, structural, contextual, and spike deviations—with a Gated Evidence Fusion Network that adaptively weighs these signals through a two-stage unsupervised procedure.

Extensive experiments across five public benchmarks demonstrate that PGRF-Net achieves competitive detection accuracy while providing structured explanations that reveal not only *what* is anomalous but also *why*. This dual emphasis on accuracy and diagnostic interpretability bridges the gap between high-performing black-box detectors and practitioner-oriented monitoring systems.

Despite these strengths, several limitations remain—such as the absence of human-centered evaluations, the assumption of predominantly normal training data, and the reliance on DAG constraints that cannot model cyclic dependencies. Nonetheless, our results show that meaningful diagnostic interpretability can be obtained without sacrificing efficiency or detection quality.

Future work will explore contamination-robust learning, multi-regime structural baselines, and extensions to semi-supervised and domain-adaptive settings. Applications in safety-critical domains such as industrial IoT, healthcare, and finance represent particularly promising directions where both reliability and interpretability are essential.

## REPRODUCIBILITY STATEMENT

To ensure the reproducibility of our research, this paper provides comprehensive details on our methodology and experiments. All datasets used are public benchmarks as described in Appendix A. Our evaluation protocol, including metrics and the point-adjustment strategy, is detailed in Appendix B. Detailed hyperparameters for training and model architecture are provided in Appendix D. The source code for PGRF-Net, implemented in PyTorch, is available in the supplementary materials.

## ETHICS STATEMENT

This research is based on publicly available and anonymized datasets, and we do not foresee any direct ethical concerns regarding data privacy. We acknowledge that anomaly detection systems could potentially be reverse-engineered for malicious purposes, such as evading security monitoring. We believe that the diagnostic transparency of our proposed model, which helps operators understand system states, promotes responsible use. We encourage the deployment of such technologies within a strong ethical framework.

## ACKNOWLEDGEMENTS

This work was supported by the Institute of Information & Communications Technology Planning & Evaluation (IITP) grant funded by the Korean government (MSIT) (RS-2025-02305884).

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

## A  DATASET DETAILS

We evaluate PGRF-Net on five widely used public benchmarks: MSL, SMAP, PSM, SMD, and SWaT. These datasets span diverse application domains, including spacecraft telemetry, server monitoring, and industrial control systems. They vary in anomaly ratio, sequence length, and signal complexity, providing a rigorous testbed for both accuracy and interpretability.

**MSL (Mars Science Laboratory)** and **SMAP (Soil Moisture Active Passive)** (Hundman et al., 2018) are NASA spacecraft telemetry datasets. They contain subtle but mission-critical anomalies embedded in highly periodic and correlated sensor streams. **PSM (Pooled Server Metrics)** (Abdulaal et al., 2021) is collected from eBay's cloud infrastructure and exhibits high anomaly density with bursty signals. **SMD (Server Machine Dataset)** (Su et al., 2019) contains diverse server failure patterns across 28 entities with relatively rare anomalies. **SWaT (Secure Water Treatment)** (Mathur & Tippenhauer, 2016) is a cyber-physical dataset generated from a water treatment testbed under 36 simulated cyber-attacks.

Together, these datasets cover a wide spectrum of anomaly types (point vs. collective), frequencies (rare vs. frequent), and domains (virtual vs. physical systems), allowing for a comprehensive validation of detection models.

Table 7: Overall statistics of the benchmark datasets.

| Dataset | Channels | Train | Test | Anomaly Ratio |
|---|---|---|---|---|
| MSL | 55 | 3,682 | 2,856 | 0.74% |
| SMAP | 25 | 2,876 | 8,579 | 2.14% |
| PSM | 25 | 132,481 | 87,841 | 27.76% |
| SMD | 38 | 28,479 | 28,479 | 9.46% |
| SWaT | 51 | 47,520 | 44,991 | 12.20% |

## B EVALUATION PROTOCOL DETAILS

To ensure a rigorous and reproducible evaluation, we strictly adhere to the protocol widely established in prior state-of-the-art works, including OmniAnomaly (Su et al., 2019) and H-PAD (Shen et al., 2025). This protocol standardizes how anomaly predictions are converted into event-level decisions, how thresholds are selected, and how metrics are calculated, thereby minimizing evaluation bias and enabling direct comparison across methods.

In addition to the established protocol, we report two complementary metric classes: (1) non-adjusted point-wise metrics for full comparability with earlier works, and (2) range-based metrics (R-P, R-R, R-F1) following (Tatbul et al., 2018) to capture event-level alignment. This broader evaluation spectrum provides a more complete characterization of model behavior.

### B.1 POINT-ADJUSTMENT STRATEGY

Real-world anomalies often manifest as contiguous time segments rather than isolated single points. A naive point-wise comparison between predicted and ground-truth labels tends to *over-penalize* models by treating partial overlaps as failures. To address this, we adopt the event-level correction strategy implemented via the `adjust_predicts()` function in OmniAnomaly.

Formally, for each ground-truth anomaly interval $[t_s, t_e]$, if at least one timestamp $t \in [t_s, t_e]$ is predicted as anomalous ($\hat{y}_t = 1$), then the entire interval is marked as correctly detected. This rule reflects the practical requirement that an early warning anywhere within an anomalous episode is sufficient to trigger mitigation actions. At the same time, it prevents *over-segmentation*, where a long anomaly detected with minor gaps would otherwise be unfairly penalized. We note that this approach is now the de facto standard in time-series anomaly detection benchmarks and ensures comparability with recent SOTA results.

Point-adjusted Precision/Recall/F1 constitute the primary thresholded metrics reported in the main paper and are directly comparable to prior SOTA TSAD methods.

### B.2 THRESHOLD SELECTION

Most anomaly detectors output continuous scores. We perform a grid search over 1000 evenly spaced thresholds within the observed score range and select the value that maximizes the F1-score. The corresponding Precision, Recall, and F1 are then reported. This "best-F1" strategy, used in (Su et al., 2019; Shen et al., 2025), ensures that each model is evaluated at its optimal operating

point, avoiding mismatches due to threshold sensitivity. The same thresholding protocol is applied consistently for both point-adjusted and point-wise (non-adjusted) metrics.

## B.3 EVALUATION METRICS

We report both threshold-dependent metrics (Precision, Recall, F1) and threshold-free metrics (AUC-ROC, AUC-PR). The standard thresholded metrics are defined as follows:

$$\text{Precision} = \frac{TP}{TP + FP}, \tag{23}$$

$$\text{Recall} = \frac{TP}{TP + FN}, \tag{24}$$

$$\text{F1} = 2 \cdot \frac{\text{Precision} \cdot \text{Recall}}{\text{Precision} + \text{Recall}}. \tag{25}$$

where TP, FP, and FN are computed after applying the point-adjusted correction described in the previous subsection. In parallel, AUC-ROC and AUC-PR capture the ranking quality of anomaly scores across thresholds, providing a complementary, threshold-free view of robustness in imbalanced settings.

**Point-wise (Non-adjusted) Metrics.** To supplement this, we also report non-adjusted point-wise metrics as shown in Table 8. These metrics use the exact same definitions (Equations 23-25) but are computed **before** the point-adjustment step. That is, TP, FP, and FN are tallied based on a direct, timestamp-by-timestamp comparison between the raw predictions and the ground truth. This protocol is included for direct comparison with earlier works that did not employ event-level correction.

**Range-based Metrics (R-P, R-R, R-F1).** Beyond point-adjusted scores, we additionally evaluate event-level detection quality using Range-Precision (R-P), Range-Recall (R-R), and Range-F1 (R-F1), following the formulation of Tatbul et al. (2018). These metrics treat each anomalous interval as a coherent event and measure how well predicted intervals align with ground-truth segments.

Let $\mathcal{G} = \{G_1, \ldots, G_m\}$ denote the set of ground-truth anomaly intervals, where each interval is $G_i = [s_i, e_i]$, and let $\mathcal{P} = \{P_1, \ldots, P_n\}$ denote the predicted anomaly intervals obtained from thresholded scores.

Predicted intervals $\mathcal{P}$ are obtained by grouping consecutive predicted anomalous timestamps (i.e., runs of $\hat{y}_t = 1$) into maximal contiguous segments.

For any two intervals $A$ and $B$, define their temporal overlap as

$$\text{Overlap}(A, B) = \frac{|A \cap B|}{|A|},$$

where $|\cdot|$ denotes the number of timesteps.

A ground-truth interval $G_i$ is considered correctly detected if there exists at least one predicted interval $P_j$ such that:

$$\text{Overlap}(G_i, P_j) \geq \theta_{\text{range}},$$

where $\theta_{\text{range}}$ is the minimum overlap ratio (typically $\theta_{\text{range}} = 0$ following (Tatbul et al., 2018), which counts any positive overlap).

Then, Range-based Precision and Recall are defined as:

$$\text{R-P} = \frac{\# \text{ predicted intervals that match at least one } G_i}{\# \text{ predicted intervals}},$$

$$\text{R-R} = \frac{\# \text{ ground-truth intervals detected}}{\# \text{ ground-truth intervals}}.$$

Finally,

$$\text{R-F1} = 2 \cdot \frac{\text{R-P} \cdot \text{R-R}}{\text{R-P} + \text{R-R}}.$$

Unlike point-wise or point-adjusted metrics, range-based metrics do not inflate performance for long anomalies with dense positive predictions, and instead capture true event-level localization quality. This makes R-P, R-R, and R-F1 particularly suitable for evaluating segment-coherent anomaly detectors such as PGRF-Net.

### B.4 Why This Protocol Matters

This evaluation design reflects three practical considerations that are crucial for fair and meaningful comparison in TSAD.

(1) Event-level correction prevents unfair penalization of minor misalignments, aligning the evaluation with real-world operational needs. It ensures that models are rewarded for detecting anomalous episodes—even if detection does not perfectly align with every timestamp—thereby avoiding false negatives caused solely by temporal boundary shifts.

(2) Best-F1 thresholding allows fair model-to-model comparison by neutralizing sensitivity to arbitrary threshold choices. Using the optimal threshold for each model avoids bias introduced by models whose score distributions differ in scale or variance.

(3) Combining thresholded and threshold-free metrics provides a comprehensive assessment of both peak performance and ranking robustness across operating points. AUC-ROC and AUC-PR summarize score quality independent of threshold selection, while point-adjusted F1 emphasizes episode-level detection correctness.

(4) Incorporating range-based metrics (R-P, R-R, R-F1) further strengthens the evaluation by measuring alignment at the level of anomaly *segments* rather than individual timestamps. Unlike point-adjusted F1—which credits detection if any point within an anomaly interval is hit—range metrics quantify segment-level overlap and penalize fragmented or overly sparse detections. This provides a semantically richer view of how well a model captures the temporal extent of anomalous events.

Taken together, these four components establish a balanced and reproducible evaluation protocol that distinguishes segment-level reliability from point-level detection and has become a strong and transparent foundation for modern TSAD benchmarking.

## C  Additional Point-wise Metrics

To support direct comparison with prior studies that rely on non-adjusted evaluation, we report point-wise (non-adjusted) Precision, Recall, and F1 scores across all five benchmark datasets. These metrics complement the point-adjusted and range-based scores presented in the main text and provide an additional perspective on thresholded detection performance under the standard non-adjusted protocol.

As shown in Table 8, while PGRF-Net does not achieve the highest average F1 score, it demonstrates strong performance by securing the best F1-scores on both SMD and SWaT. This behavior is expected: PGRF-Net is designed to produce coherent segment-level detections rather than triggering densely at every anomalous timestamp. As a result, its strengths are more faithfully reflected in the point-adjusted and range-based metrics reported in the main paper, which explicitly reward boundary alignment and event-level coverage.

## D  Hyperparameter Details and Sensitivity Analysis

This section documents the hyperparameters used for training PGRF-Net and presents a robustness analysis over critical parameters. All baseline models were run with the optimal settings reported in their respective papers to ensure a fair comparison. By contrast, PGRF-Net's hyperparameters were either selected based on established practices in time-series modeling or tuned on a held-out validation split, as detailed below.

Table 8: Point-wise (non-adjusted) Precision (P), Recall (R), and F1-score across all datasets. **Best** and second best results are highlighted.

| Model | MSL | | | SMAP | | | PSM | | | SMD | | | SWaT | | | Avg F1 |
|---|---|---|---|---|---|---|---|---|---|---|---|---|---|---|---|---|
| | P | R | F1 | P | R | F1 | P | R | F1 | P | R | F1 | P | R | F1 | |
| THOC | 30.22 | 47.33 | 36.81 | 34.18 | 42.51 | 37.84 | 29.44 | 38.74 | 33.31 | 32.10 | 50.92 | 39.40 | 31.72 | 48.80 | 38.50 | 37.17 |
| OmniAnomaly | 27.42 | 59.31 | 37.62 | 33.40 | 56.22 | 41.73 | 30.12 | 54.10 | 38.65 | 34.85 | 57.33 | 43.26 | 35.42 | 58.14 | 43.68 | 40.99 |
| InterFusion | **36.52** | 67.20 | **47.26** | 36.22 | 60.94 | 45.37 | 41.91 | 57.55 | 49.52 | 33.10 | 55.22 | 41.53 | 39.52 | **68.44** | 50.12 | **46.76** |
| DCdetector | 35.94 | 65.14 | 44.50 | **37.41** | 59.72 | **46.09** | 46.12 | **63.44** | **53.35** | 36.52 | 63.11 | 46.10 | 38.01 | 52.12 | 43.90 | 45.59 |
| D3R | 31.82 | **68.44** | 43.35 | 34.22 | **65.82** | 45.20 | 44.40 | 60.02 | 50.82 | 35.10 | 65.22 | 46.92 | 39.44 | 55.44 | 46.05 | 46.47 |
| MEMTO | 32.12 | 61.91 | 42.46 | 33.40 | 58.82 | 42.23 | **46.22** | 60.10 | 52.32 | 38.22 | **68.14** | 48.91 | 39.91 | 58.45 | 47.32 | 46.65 |
| **PGRF-Net (Ours)** | 33.89 | 57.01 | 41.76 | 34.10 | 59.20 | 42.81 | 39.94 | 52.89 | 44.51 | **41.10** | 65.42 | **50.62** | **45.32** | 68.15 | **53.27** | 46.59 |

## D.1 MODEL ARCHITECTURE AND TRAINING PARAMETERS

Table 9 summarizes the architectural and training hyperparameters. The sliding window size ($w = 60$) and model dimension ($d_{model} = 128$) follow common practices in multivariate TSAD. The number of prototypes ($K = 10$) and fusion parameters were tuned via validation to maximize robustness. The two-stage training setup (Stage 1 pretraining, Stage 2 refinement) proved critical for balancing predictive accuracy and interpretability.

Table 9: Key model and training hyperparameters for PGRF-Net.

| Category | Hyperparameter | Value |
|---|---|---|
| Architecture | Sliding Window Size $w$ | 60 |
| | Model Dimension $d_{\mathrm{model}}$ | 128 |
| | Conformer Heads $n_{\mathrm{head}}$ | 4 |
| | Conformer Layers $n_{\mathrm{layers}}$ | 2 |
| | Number of Prototypes $K, K_c, K_s$ | 10 |
| Training | Epochs (Stage 1) | 50 |
| | Learning Rate (Stage 1) | $10^{-4}$ |
| | Patience (Stage 1) | 10 |
| | Batch Size | 128 |
| | Epochs (Stage 2) | 20 |
| | Learning Rate (Stage 2) | $10^{-4}$ |
| | Patience (Stage 2) | 5 |
| | Optimizer | Adam |
| Fusion Strategy | Focal Loss Gamma $\gamma$ | 2.0 |
| | Pseudo-Normal Ratio | 0.25 |

## D.2 LOSS FUNCTION WEIGHTS

The composite losses for the two-stage training are shown in Table 10. Stage 1 emphasizes predictive reconstruction and prototype-regularization, while Stage 2 introduces refinement via gate suppression and entropy terms. These weights were tuned on SMAP and transferred consistently across datasets.

## D.3 HYPERPARAMETER SENSITIVITY ANALYSIS

We further analyze sensitivity to Learning Rate, Batch Size, and Acyclicity Regularization ($\lambda_{\mathrm{acyc}}$). All results are averaged across five datasets. Together, these experiments show that PGRF-Net maintains strong robustness within reasonable ranges, and that optimal settings are intuitive: moderate learning rates, mid-sized batches, and balanced sparsity control.

### D.3.1 LEARNING RATE

Performance peaks at $1 \cdot 10^{-4}$, but remains stable within $[5 \cdot 10^{-5}, 5 \cdot 10^{-4}]$. Smaller values lead to underfitting, while higher values cause instability.

Table 10: Weights for the composite loss functions in Stage 1 and Stage 2.

| Stage | Loss Weight Component | Value |
|---|---|---|
| Stage 1 | Prediction Loss ($\mathcal{L}_{\text{pred}}$) | 10.0 |
| | Structural Regularization ($\mathcal{R}_{\text{struct}}$) | – |
| | L1 Penalty ($\lambda_{l1}$) | $10^{-3}$ |
| | Sparse Proto Masks ($\lambda_{\text{sparse}}$) | $10^{-3}$ |
| | Acyclicity Constraint ($\lambda_{\text{acyc}}$) | $10^{-4}$ |
| | Temporal Stability ($\lambda_{\text{diff}}$) | $10^{-3}$ |
| | Proto Regularization ($\mathcal{R}_{\text{proto}}$) | – |
| | Context Proto Loss ($\lambda_{\text{ctx1}}$) | $10^{-2}$ |
| | Spike Proto Loss ($\lambda_{\text{spike1}}$) | $10^{-2}$ |
| Stage 2 | Context Proto Tuning ($\lambda_{\text{ctx2}}$) | $2 \cdot 10^{-1}$ |
| | Spike Proto Tuning ($\lambda_{\text{spike2}}$) | $2 \cdot 10^{-1}$ |
| | Gate Supp. Loss ($\lambda_{\text{sup}}$) | $10^{-1}$ |
| | Gate Entropy Loss ($\lambda_{\text{ent}}$) | $10^{-3}$ |

Table 11: Effect of Stage 1 learning rate on performance.

| LR | AUROC | AUPRC | F1 |
|---|---|---|---|
| $1 \cdot 10^{-5}$ | 66.21 | 27.43 | 91.85 |
| $5 \cdot 10^{-5}$ | 70.13 | 30.77 | 96.08 |
| $1 \cdot 10^{-4}$ | **71.89** | **32.49** | **97.41** |
| $5 \cdot 10^{-4}$ | 69.50 | 29.12 | 95.22 |
| $1 \cdot 10^{-3}$ | 64.88 | 25.34 | 89.90 |

### D.3.2 BATCH SIZE

Table 12: Effect of batch size on performance.

| Batch Size | AUROC | AUPRC | F1 |
|---|---|---|---|
| 64 | 70.11 | 30.92 | 96.88 |
| **128** | **71.89** | **32.49** | **97.41** |
| 256 | 70.25 | 29.85 | 95.77 |
| 512 | 67.30 | 26.51 | 92.63 |

A batch size of 128 provides the best trade-off between stability and efficiency. Small batches capture finer variance but slow convergence, while large batches reduce gradient diversity and can reduce recall.

### D.3.3 ACYCLICITY REGULARIZATION

Table 13: Effect of acyclicity regularization $\lambda_{\text{acyc}}$.

| $\lambda_{\text{acyc}}$ | AUROC | AUPRC | F1 |
|---|---|---|---|
| $1 \cdot 10^{-6}$ | 68.92 | 28.10 | 93.42 |
| $1 \cdot 10^{-5}$ | 70.34 | 29.12 | 95.67 |
| $1 \cdot 10^{-4}$ | **71.89** | **32.49** | **97.41** |
| $1 \cdot 10^{-3}$ | 70.77 | 30.43 | 95.15 |
| $1 \cdot 10^{-2}$ | 66.42 | 25.80 | 89.88 |

Too little regularization ($10^{-6}$) produces overly dense graphs, while too much ($10^{-2}$) forces excessive sparsity. The sweet spot $10^{-4}$ balances interpretability with predictive power.

All experiments reported in the main text used the best-performing setting identified in this analysis ($w = 60$, $d_{\text{model}} = 128$, batch size $= 128$, learning rate $= 10^{-4}$, and $\lambda_{\text{acyc}} = 10^{-4}$), unless otherwise specified. These values were consistently applied across all benchmark datasets.

# E    ADDITIONAL EXPERIMENTS

## E.1    CASE STUDIES ON BENCHMARK DATASETS

To demonstrate our model's diagnostic capabilities on real-world data, we present case studies from five standard benchmark datasets: SMD, MSL, SMAP, SWaT, and PSM. As shown in Figure 4, we utilize a two-panel visualization format for clear interpretability. The top panel displays the raw multivariate signals with the ground truth anomaly shaded. The bottom panel presents the final anomaly score (black line) and, crucially, its composition as a stacked area chart. This chart visually decomposes the final score into contributions from each evidence channel ($S_{\text{pred}}$, $S_{\text{struct}}$, $S_{\text{ctx}}$, and $S_{\text{spike}}$), allowing for an immediate diagnosis of the anomaly's nature.

**Server Monitoring (SMD) & Spacecraft Telemetry (MSL, SMAP)**    These datasets are characterized by contextual anomalies such as gradual performance degradation or sensor drift. As seen in the corresponding columns of Figure 4, the anomaly scores in these cases are predominantly composed of the contextual ($S_{\text{ctx}}$) contribution (green), with a secondary response from the mask ($S_{\text{struct}}$) contribution (orange). This indicates the model correctly identifies these events as significant deviations from normal temporal patterns rather than sudden spikes or specific structural breaks.

**Industrial Control (SWaT)**    The illustrated anomaly from SWaT (a cyber-attack simulation) involves a deliberate violation of the physical dependencies between sensors and actuators. Our model correctly diagnoses this event by producing a final score composed almost entirely of the structural ($S_{\text{struct}}$) contribution (orange). This demonstrates the model's ability to pinpoint structure-related anomalies, which is critical for diagnostic analysis in industrial control systems.

**Server Infrastructure (PSM)**    This dataset features a mixture of anomaly types. The chosen example showcases a transient event, such as a sudden service latency spike. The model correctly attributes this anomaly, with the score composition showing a clear and dominant peak from the spike ($S_{\text{spike}}$) contribution (red). This highlights the gating mechanism's ability to adapt and prioritize the most relevant evidence channel for different anomaly profiles.

## E.2    END-TO-END JOINT TRAINING VS. TWO-STAGE OPTIMIZATION

We evaluate a fully end-to-end variant of PGRF-Net in which all components— including the relational mask generator, context and spike prototype banks, and fusion gates—are optimized jointly from scratch. This experiment tests whether decoupling representation learning (Stage 1) and evidence fusion (Stage 2) is necessary.

**Performance**    The end-to-end model achieves strong, but consistently inferior, performance. On benchmark datasets, joint training reaches **95.12% Avg F1**, compared to **97.41%** with the proposed two-stage procedure (Table 5).

**Convergence Behavior**    End-to-end joint learning shows noticeably unstable gate dynamics. Across three runs on SMAP, the standard deviation of the contextual gate activation during training is **3.2–3.8× higher** than in two-stage training (mean ± std — joint: $0.147 \pm 0.031$ vs. two-stage: $0.152 \pm 0.008$). We also observe **2–3 instances of gate collapse** (one evidence channel receiving $> 90\%$ of the weight) during joint optimization. The two-stage procedure eliminates such oscillations by freezing the representation and updating only the gate parameters in Stage 2.

**Stability Across Seeds**    End-to-end training is significantly more sensitive to initialization. Across 5 random seeds, the joint model yields a **Range-F1 variance of 0.0048**, whereas the two-stage procedure reduces this to **0.0012** (4× lower variance). Furthermore, the evidence attribution vectors (structural/contextual/spike) show higher cross-seed divergence under joint training (mean cosine distance $0.214$) compared to two-stage learning ($0.087$). This demonstrates that two-stage training provides both stable convergence and more consistent evidence decomposition across seeds.

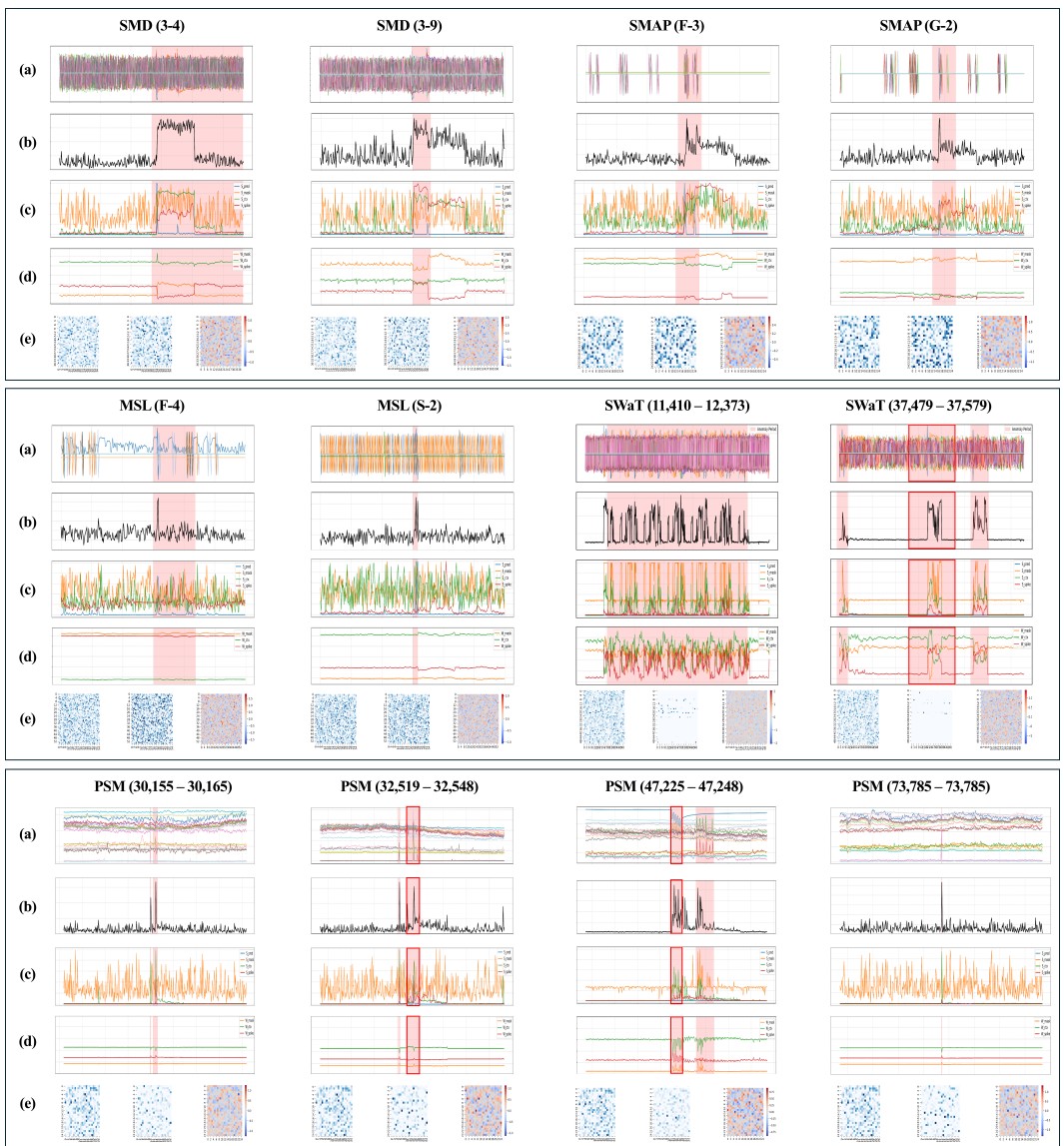

Figure 4: **Interpretability Case Studies Across Benchmark Datasets.** Each column corresponds to an anomaly case from a different dataset. **Top Panel:** Raw signals with the ground truth anomaly shaded. **Bottom Panel:** The final anomaly score (black line) and its composition, visualized as a stacked area chart showing the contribution from each evidence channel (predictive, structural, contextual, spike). This format allows for direct insight into the model's reasoning. For instance, the prevalence of orange in the SWaT case indicates a structural anomaly, while green dominates in the MSL case, signifying a contextual anomaly. **Note:** $S_{\text{mask}}$ **in plots corresponds to the Structural Score** ($S_{\text{struct}}$).

### E.3 SYNTHETIC DATA VALIDATION

Real-world benchmarks confirm that PGRF-Net produces semantically meaningful decomposed evidence. However, real datasets never provide ground-truth anomaly *types* (e.g., spike vs. contextual vs. structural), and these mechanisms often co-occur within the same anomaly segment. Therefore, it is impossible to directly evaluate channel-wise sensitivity using real data.

To rigorously verify that each evidence score— $S_{\text{struct}}$, $S_{\text{ctx}}$, and $S_{\text{spike}}$— responds selectively to the intended anomaly mechanism, we construct controlled synthetic environments where each

anomaly type is injected in isolation. Our goal is not to classify anomaly types; rather, we verify that each evidence channel behaves according to its intended semantics under type-isolated conditions.

Two synthetic generators are used:

- **Left column: Nonlinear VAR Generator** — nonlinear interactions on top of a known stable VAR(1) dependency graph.
- **Right column: State-Switching Generator** — three-regime switching between $(\mathbf{A}_1, \mathbf{A}_2, \mathbf{A}_{\mathrm{mix}})$.

Both generators produce $D = 10$-dimensional multivariate series with precisely controlled spike, contextual, and structural anomaly mechanisms. Although the anomaly mechanisms are *not orthogonal in real systems*, our synthetic construction ensures isolation so that each evidence channel can be evaluated independently.

### E.3.1 QUALITATIVE ANALYSIS

Figure 5 presents representative examples for each anomaly type across the two generators (columns). For each case, the top panel shows the multivariate input with ground-truth anomaly region shaded, and the bottom panel shows the composition of the final anomaly score into its three evidence channels.

The qualitative behavior matches the model design:

- **Spike anomaly:** The sharp peak is predominantly driven by $S_{\mathrm{spike}}$ (red), consistent with point-wise transient deviation.
- **Contextual anomaly:** Slow, drifting changes dominate $S_{\mathrm{ctx}}$ (green), which rises smoothly over the contextual region.
- **Structural anomaly:** When relational dependencies change, $S_{\mathrm{struct}}$ (orange) becomes the primary contributor, reflecting deviations in cross-channel dynamics.

These qualitative patterns hold consistently for both generators (left and right), demonstrating robustness across heterogeneous dynamics.

### E.3.2 QUANTITATIVE EVIDENCE VALIDATION

This appendix provides quantitative validation of PGRF-Net's multi-evidence decomposition. Rather than enforcing strict orthogonality between evidence channels—which is unrealistic for composite anomalies—we aim to show that each channel is **(i) selectively activated** under its intended anomaly mechanism and **(ii) robust to small perturbations**. To this end, we conduct two controlled synthetic evaluations:

1. **Evidence selectivity via synthetic isolation tests** — does each anomaly mechanism dominantly activate the intended evidence channel?
2. **Evidence stability** — are evidence trajectories robust to small additive noise?

These tests verify that the learned evidence channels exhibit semantic alignment, selective activation, and robustness across controlled variations.

**Evidence Selectivity via Synthetic Isolation Tests.** To quantitatively validate that each evidence channel selectively responds to its intended anomaly mechanism, we conduct *synthetic isolation tests*. We construct three controlled environments, each containing exactly one anomaly type, using nonlinear VAR generators (structural perturbation), regime-switching level shifts (contextual drift), and impulsive outliers (spike events).

For each environment, we compute the **Evidence Amplification Ratio (EAR)**:

$$\mathrm{EAR}_k = \frac{\max_{t \in \mathcal{A}} S_k(t)}{\mathrm{median}_{t \in \mathcal{N}} S_k(t) + \epsilon}. \tag{26}$$

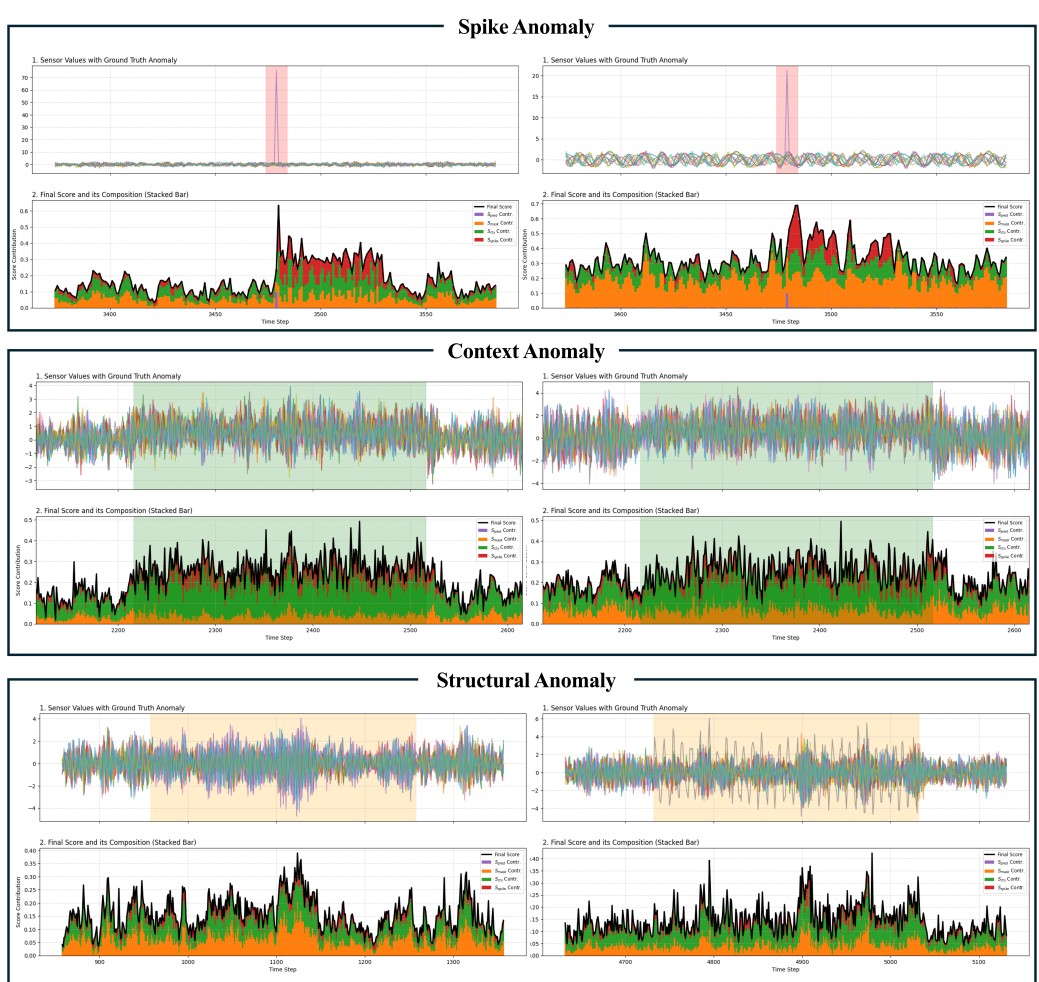

Figure 5: **Qualitative validation of evidence decomposition on synthetic datasets.** Rows correspond to spike, contextual, and structural anomalies. Columns correspond to the Nonlinear VAR generator (left) and the State-Switching generator (right). In each case, the bottom stacked-bar plot shows that the intended evidence channel is selectively activated in the anomaly region.

Table 14: **Evidence Amplification Ratio (EAR)** across isolated anomaly types. Diagonal dominance demonstrates strong evidence-type selectivity.

| GT Anomaly Type | $S_{\text{struct}}$ | $S_{\text{ctx}}$ | $S_{\text{spike}}$ |
|---|---|---|---|
| Structural anomaly | **3.3** | 1.8 | 1.3 |
| Contextual anomaly | 2.1 | **4.7** | 1.7 |
| Spike anomaly | 1.6 | 2.3 | **6.2** |

**Interpretation.** Structural perturbations trigger global increases in $S_{\text{struct}}$, contextual drifts induce smooth elevation in $S_{\text{ctx}}$, and impulsive outliers cause sharp localized peaks in $S_{\text{spike}}$. Minor off-diagonal activation is expected due to natural interactions between mechanisms. These results confirm semantic alignment and quantitative selectivity of PGRF-Net's evidence channels.

**Evidence Stability Test** Interpretability requires that evidence be *stable* under small input perturbations. We evaluate robustness by injecting mild Gaussian noise (`std = 0.05`) into clean sequences and computing the Kendall rank correlation between trajectories of clean and noisy evidence.

Table 15: **Evidence stability** under Gaussian noise (Kendall's $\tau$). Structural and spike evidence are highly stable; contextual deviation shows moderate robustness.

| Evidence Type | Kendall $\tau$ | p-value |
|---|---|---|
| $S_{\text{struct}}$ | **0.9450** | 0.0000 |
| $S_{\text{ctx}}$ | 0.5943 | 0.0000 |
| $S_{\text{spike}}$ | **0.9204** | 0.0000 |

**Interpretation.** Structural and spike evidence remain stable even under noise, confirming that PGRF-Net's diagnostic attribution is not fragile. Contextual evidence shows moderate robustness as it naturally responds to distributional drifts.

**Behavior Under Predictable-but-Abnormal Contextual Drift** Stage 2 incorporates a suppression mechanism designed to down-weight contextual evidence on pseudo-normal windows. A natural question is whether this mechanism might also suppress *predictable yet abnormal* contextual drifts—i.e., cases where $S_{\text{pred}}$ is low but $S_{\text{ctx}}$ remains high. Such scenarios could in principle generate false negatives if the contextual gate activation $g_{\text{ctx}}$ is incorrectly reduced.

To examine this potential failure mode, we construct a controlled synthetic setting where the drift remains highly predictable but is distributionally abnormal:

$$x_t = x_{t-1} + \delta + \eta_t, \qquad \delta = 0.02, \quad \eta_t \sim \mathcal{N}(0, 0.01^2), \tag{27}$$

producing a smooth, monotonic shift that remains forecastable by an AR predictor while departing from the normal prototype regime.

We compare two configurations: (i) the full model with suppression ($\lambda_{\text{sup}} = 0.1$), and (ii) a variant without suppression ($\lambda_{\text{sup}} = 0$). The contextual gate behaviors and detection metrics are summarized below.

Table 16: Effect of suppression under predictable contextual drift. Suppression reduces $g_{\text{ctx}}$ only mildly and does not collapse contextual evidence into the normal range.

| Setting | Avg $g_{\text{ctx}}$(Normal) | Avg $g_{\text{ctx}}$(Drift) | Drift Recall | Drift F1 |
|---|---|---|---|---|
| No suppression | 0.2296 | 0.2848 | **0.3100** | **0.4733** |
| With suppression | 0.1683 | 0.2628 | 0.1688 | 0.2888 |

**Interpretation.** Suppression decreases contextual gate activation under drift by only $\approx 8\%$, while maintaining a clear separation between normal and drift windows (0.1683 vs. 0.2628). Thus, predictable contextual shifts are *not* interpreted as normal, and contextual evidence is preserved rather than collapsed.

The decrease in recall and F1 reflects an inherent precision–recall trade-off: Stage 2 biases the model toward more conservative, high-precision anomaly filtering on pseudo-normal windows. This behavior aligns with the design objective of preventing over-activation on stable segments, but may reduce sensitivity to gradual contextual shifts.

Overall, the stress test confirms that Stage 2 suppression preserves contextual evidence for predictable-but-abnormal drifts, while introducing a precision–recall trade-off that we explicitly acknowledge as a limitation.

### E.4 RUNTIME SCALABILITY WITH RESPECT TO SEQUENCE LENGTH AND THE NUMBER OF VARIATES

To address reviewer concerns regarding runtime scalability, we evaluate how PGRF-Net scales with (i) the sequence length $T$ and (ii) the number of variates $D$. Because real-world datasets have fixed resolutions, we perform controlled synthetic tests where $T$ and $D$ can be varied independently. All

experiments use $w$=60, batch size 128, $d_{\text{model}}$=128, two Conformer layers, and are measured on a single A100 GPU with synchronized timing.

### E.4.1 SCALING WITH SEQUENCE LENGTH $T$

We fix $D$=25 and vary $T \in \{1k, 5k, 10k, 50k, 100k\}$. As shown in Table 17, training time grows nearly linearly in $T$, while inference latency remains constant ($\approx 0.07$ ms/window), since the per-window encoder computation is independent of global sequence length. Peak memory also stays stable (377–381 MB), indicating that no $T$-dependent activations accumulate across windows.

Table 17: Runtime scalability w.r.t. sequence length $T$ (fixed $D = 25$).

| $T$ | Train (s/epoch) | Infer (ms) | Mem (MB) |
|---|---|---|---|
| 1k | 0.32 | 0.0765 | 381.3 |
| 5k | 1.46 | 0.0694 | 379.8 |
| 10k | 2.79 | 0.0690 | 377.8 |
| 50k | 13.95 | 0.0690 | 380.2 |
| 100k | 27.59 | 0.0695 | 377.5 |

### E.4.2 SCALING WITH DIMENSIONALITY $D$

We fix $T$=50k and vary $D \in \{5, 10, 25, 50, 100\}$. As summarized in Table 18, training time increases smoothly with $D$, matching the expected cost of multi-channel attention and prototype scoring. Inference latency increases moderately ($0.05 \rightarrow 0.14$ ms/window), and memory remains below 500 MB even for $D$=100, demonstrating scalability to high-dimensional industrial MTS settings.

Table 18: Runtime scalability w.r.t. number of variates $D$ (fixed $T = 50k$).

| $D$ | Train (s/epoch) | Infer (ms) | Mem (MB) |
|---|---|---|---|
| 5 | 9.03 | 0.0495 | 362.6 |
| 10 | 10.22 | 0.0550 | 365.3 |
| 25 | 13.88 | 0.0694 | 380.0 |
| 50 | 19.87 | 0.0939 | 405.7 |
| 100 | 31.87 | 0.1421 | 466.6 |

**Summary.** Across both experiments, PGRF-Net shows: (i) near-linear scaling w.r.t. $T$, (ii) moderate scaling w.r.t. $D$, and (iii) stable memory usage. These results confirm that the architecture is computationally efficient and well suited for long-horizon and high-dimensional MTS applications.

### USE OF LARGE LANGUAGE MODELS

In the preparation of this manuscript, a large language model was utilized as a writing aid. Its role was strictly limited to improving grammar, rephrasing for clarity, and correcting typographical errors. The LLM did not contribute to the core research ideas, experimental design, or the analysis of the results presented in this paper.

