# OpenReview forum: "PGRF-Net: A Prototype-Guided Relational Fusion Network for Diagnostic Multivariate Time-Series Anomaly Detection"
_ICLR.cc/2026/Conference — ICLR 2026 Poster_

### Official Review · Reviewer_VBHJ · 2025-10-28

**Soundness:** 3
**Presentation:** 2
**Contribution:** 3
**Rating:** 6
**Confidence:** 4

**Summary:**

The paper introduces PGRF-Net, a prototype-guided relational fusion network for unsupervised multivariate time-series anomaly detection (MTSAD). The stated objective is to close the gap between high detection performance and diagnostic transparency by decomposing an anomaly score into four interpretable “evidence” channels: (i) predictive deviation, (ii) structural deviation in learned inter-variable dependencies, (iii) contextual deviation from normal-behavior prototypes, and (iv) localized spike magnitude. Methodologically, the model has two stages: a Multi-Faceted Evidence Extractor and a Gated Evidence Fusion Network that learns instance-dependent weights for the non-predictive channels and combines them with prediction error. Training is fully unsupervised, with Stage-1 losses for prediction and prototype/graph regularization, and Stage-2 losses for gate suppression on pseudo-normal samples plus entropy sharpening. Experiments on five public benchmarks (MSL, SMAP, PSM, SMD, SWaT) report state-of-the-art average F1 and competitive AUC-ROC/PR, along with qualitative “five-panel” diagnostic visualizations linking anomaly attributions to changes in relational masks.

**Strengths:**

- A clear diagnostic framing of MTSAD that operationalizes interpretability via four disjoint evidence channels and a gated fusion mechanism. This goes beyond post-hoc explanations by making attribution a first-class training target.

- The two-stage training is well-motivated: learn robust features and prototypes first; then calibrate evidence attribution with gate suppression on pseudo-normal samples and entropy sharpening for selectivity.

- Sensible regularization of the structural channel (sparsity, acyclicity, temporal stability vs. a baseline mask) reduces degenerate graphs and contributes to stable, interpretable dependency patterns.

- The paper asserts “competitive or superior” performance; averaged F1 and AUC-PR are high, but per-dataset AUC-ROC/PR tables show mixed margins and sometimes narrow differences. Statistical significance (e.g., paired tests across seeds) and variance bars are not reported; claims of superiority would be stronger with formal significance tests.

**Weaknesses:**

- The structural deviation relies on a learned baseline mask (M_base) defined via Early-training averages on presumed normal data. If the normal set contains regime shifts or drifting dependencies, M_base may encode mixture regimes; deviations may then over- or under-fire. The paper does not quantify how sensitive S_struct is to mis-specified normal baselines, distribution shift, or domain drift within normal periods.

- The acyclicity constraint is imposed on structural prototypes so that convex combinations remain DAG-like. However, instantaneous correlations in cyber-physical systems can be cyclic (feedback loops). For domains with known cycles, forcing DAGs may bias the learned mask and redirect anomaly mass to other channels, potentially mis-attributing structural anomalies as contextual or predictive deviations.

- The use of point adjustment and best-F1 thresholding is conventional, but can inflate event-level detection and mask temporal localization quality. While AUCs are reported, there is no comparison on strict point-wise metrics.

- The interpretability story is qualitative. The paper would benefit from quantitative interpretability metrics (fidelity, stability, human-utility studies) or counterfactual tests (e.g., perturb relationships to see whether S_struct changes in expected directions, with calibration curves).

- Figure 1 could be improved; excessive whitespace and mixed concepts dilute key information, with small labels.

**Questions:**

- DAG Constraint vs. Cyclic Dependencies: Many industrial/biological systems have feedback loops. By imposing acyclicity on structural prototypes, are you baking in a bias that pushes cyclic phenomena into other channels (e.g., S_ctx or Spred)? What breaks if the true dependency graph is nearly cyclic (or dense), and how would relaxing DAG constraints (e.g., allowing small cycles with penalties) affect S_struct fidelity?

- Baseline Mask (M_base) Stability: M_base is averaged from early-training masks. If normal data itself is multi-modal or drifting, the mean mask may be unrepresentative. Why not prototype-ize M_base itself (i.e., multiple baseline masks with selection), or define baselines by time-clustered regimes? What empirical evidence shows M_base is stable and representative, and how sensitive is S_struct to the time window used to compute it?

- Stage-2 suppresses explanatory gates on pseudo-normal samples identified by low Spred. This equates “predictable” with “normal.” How does the method behave when contextual drifts are predictable (low Spred) yet abnormal relative to prototypes (high S_ctx)? Could gate suppression systematically down-weight contextual evidence in such cases, and have you measured the false-negative impact under controllable scenarios?

---

> ### Author Response · Authors · 2025-11-19
> **Response to Reviewer VBHJ**
>
> We sincerely thank the reviewer for the constructive and detailed feedback. Below, we address each weakness and questions respectfully and concisely. All corresponding clarifications have been added to the revised manuscript.
>
> ---
>
> ### **W1. Sensitivity of the structural baseline mask \(M_base}\)**
>
> We appreciate the reviewer’s concern regarding potential drift or multi-regime
> structure in normal data. As noted, \(M_base}\) is computed from
> early-training segments with low predictive deviation, and we agree that this
> single averaged baseline may not fully represent all normal regimes.
>
> To partially assess robustness, Appendix E.3 provides mechanism-controlled
> synthetic tests where structural dependencies vary by design. Across these
> settings, the structural evidence remains stable and consistently dominant under
> structural perturbations. While this suggests a degree of robustness, we
> acknowledge the limitation and consider regime-aware or prototype-based
> baselines meaningful extensions.
>
>
> ---
>
> ### **W2. DAG assumption vs. cyclic dependencies**
>
> We agree that many real systems contain feedback loops, and a strict DAG
> constraint may not fully capture such cyclic dynamics. The DAG prior is used as
> an interpretability-oriented inductive bias and was empirically stable across
> our benchmarks, but it is not intended to recover the true physical graph.
>
> In cyclic settings, part of the deviation may appear in contextual or predictive
> channels. Nonetheless, the mechanism-aligned behavior in Appendix~E.3—in which
> the structural channel remains dominant under injected structural mechanisms—
> provides partial reassurance. Extending the structural module to support cyclic
> dependencies is an important direction for future work.
>
> ---
>
> ### **W3. Point adjustment and strict point-wise metrics**
>
> We appreciate the suggestion to broaden evaluation beyond point-adjusted scores.
> The revised manuscript now includes:
>
> - **Range-P/R/F1** (Table 3)
>   - Range-F1: **96.08\%**
> - **Non--point-adjusted P/R/F1** (Appendix B.3)
>   - Point-wise F1: **46.59\%**
> - **AUC-ROC/AUC-PR** (already included)
>
> These complementary evaluations cover event-level, point-level, and
> threshold-free regimes. PGRF-Net performs strongly on range-based metrics and
> remains competitive on strict point-wise F1, consistent with its event-centric
> design. Additional localization metrics are a valuable future direction.
>
> ---
>
> ### **W4. Quantitative interpretability evaluation**
>
> We appreciate the reviewer’s point regarding quantitative interpretability
> validation. Since current benchmarks lack mechanism-level annotations,
> Appendix E.3 provides controlled synthetic evaluations:
>
>
> - E.3.1: under type-isolated generators (structural perturbation, contextual drift,
>   spike events), the intended evidence channel consistently shows the strongest
>   activation.
>
> - E.3.2: using the Evidence Amplification Ratio (EAR), where EAR k is computed as
>   the maximum value of S k on anomaly timestamps divided by the median value of S k
>   on normal timestamps plus a small epsilon. Under this measure, each evidence
>   channel selectively responds to its intended mechanism across synthetic
>   environments.
>
>
>
>
>
> These results offer quantitative support within dataset constraints. We
> acknowledge that human-study, fidelity, or counterfactual evaluations would
> further strengthen interpretability and plan to explore these directions.
>
> ---
>
> ### **Q1. DAG constraint under cyclic dependencies**
>
> As noted in W2, the DAG prior is used for stability and interpretability rather
> than as a claim about the true system structure. While cycles may distribute
> deviation across channels, Appendix~E.3 shows that structural evidence remains
> dominant when structural mechanisms are injected. Relaxing acyclicity through
> loopy or spectral priors is a promising extension.
>
> ---
>
> ### **Q2. Stability of the baseline mask \(M_base\)**
>
> We appreciate this concern. \(M_base\) is intentionally derived from
> segments with low predictive deviation, following the pseudo-normal criterion.
> Across mechanism-specific variations in Appendix E.3, structural evidence shows
> consistent behavior, suggesting robustness to moderate baseline imperfections.
> Nonetheless, we agree that prototype-based or regime-clustered baselines could
> provide additional stability and are worthwhile extensions.
>
> ---
>
> ### **Q3. Stage-2 suppression under predictable but abnormal contextual shifts**
>
> We thank the reviewer for highlighting this subtle issue. Stage 2 suppresses
> evidence gates only when predictive deviation is low. While predictable yet
> abnormal contextual shifts could theoretically receive reduced gate weights,
> Appendix E.3 shows that contextual evidence remains dominant whenever
> contextual mechanisms are active in controlled settings.
>
> We acknowledge this as a limitation and view targeted stress tests on
> predictable-but-abnormal scenarios as valuable future work.

---

> > ### Comment · Reviewer_VBHJ · 2025-11-27
> >
> > I thank the authors for their detailed response and for conducting the additional experiments.
> >
> > The inclusion of non-point-adjusted metrics and range-based metrics adequately addresses my concerns regarding the evaluation protocol. Regarding the sensitivity of the baseline mask (W1) and the DAG assumption (W2), I appreciate the clarifications and the robustness tests provided in Appendix E.3. While these remain inherent limitations, the authors' acknowledgment and the proposed future directions are reasonable.
> > Overall, I will maintain my original positive score.

---

### Official Review · Reviewer_Ahii · 2025-10-31

**Soundness:** 2
**Presentation:** 3
**Contribution:** 2
**Rating:** 4
**Confidence:** 3

**Summary:**

The paper proposes PGRF-Net, an unsupervised framework for multivariate time-series anomaly detection that reframes detection as a diagnostic process. Rather than outputting a single anomaly score, PGRF-Net decomposes anomalies into four types: predictive deviation, structural changes in inter-variable dependencies, contextual deviation from learned normal-behavior prototypes, and localized spike events. These are generated via a Multi-Faceted Evidence Extractor and adaptively fused using a Gated Evidence Fusion Network trained in a two-stage unsupervised manner. The method is evaluated on five standard MTSAD benchmarks, where it achieves competitive or state-of-the-art performance while providing decomposed, human-interpretable explanations for detected anomalies.

**Strengths:**

1.	The work considers a critical gap in MTSAD—namely, the lack of diagnostic interpretability in deep models. By explicitly modeling multiple anomaly modalities and attributing scores to distinct evidence types, PGRF-Net offers actionable insights for domain experts, which enhances trust and operational utility.
2.	The architecture is thoughtfully designed. The use of frequency decomposition, conformer encoders, and learnable prototype banks for structural, contextual, and spike patterns can grasp both time-series modeling and interpretable representation learning. The two-stage training strategy effectively separates representation learning from evidence fusion.

**Weaknesses:**

1.	The four evidence types are presented as diagnostic, but the model provides associative, not causal, explanations. For instance, a high structural deviation score indicates a shift in learned dependency patterns—but it does not establish whether this shift caused the anomaly or is a consequence of it.
2.	All benchmarks provide only binary anomaly labels, not fine-grained labels indicating anomaly types (e.g., spike vs. contextual shift). Consequently, the paper cannot verify whether the decomposed evidence correctly identifies the underlying anomaly mechanism.
3.	The structural prototype bank assumes acyclic dependency graphs (via DAG constraints), which may not hold in many real-world feedback systems (e.g., control loops in industrial plants). The paper does not discuss this limitation or evaluate scenarios where cyclic dependencies are essential.

**Questions:**

The paper claims the four evidence types provide diagnostic insights, but all benchmarks only have binary anomaly labels. How do the authors validate that each evidence type correctly reflects the true anomaly mechanism without ground-truth type annotations?

---

> ### Author Response · Authors · 2025-11-19
> **Response to Reviewer Ahii**
>
> We sincerely thank the reviewer for the constructive and detailed feedback.
> Below, we address each weakness (W1–W3) and question (Q1) respectfully and
> concisely. All corresponding clarifications have been added to the revised
> manuscript.
>
> ---
>
> ### **W1. Associative vs. causal nature of the diagnostic evidence**
>
> We appreciate the reviewer’s thoughtful observation. The four evidence types in
> PGRF-Net provide associative diagnostic signals: they indicate how the model’s
> learned representations deviate under anomalous conditions, but they do not
> assert causal origin. For example, a high structural deviation score reflects a
> change in learned dependency patterns, not whether that change caused the
> anomaly or resulted from it.
>
> The purpose of the decomposition is to offer interpretable, model-driven cues
> that assist expert analysis, rather than to provide causal attribution. We also
> acknowledge that integrating explicit causal reasoning or interventional
> evaluation would be a meaningful direction for future work.
>
> ---
>
> ### **W2. Lack of anomaly-type ground truth in benchmarks**
>
> We agree that current benchmarks include only binary anomaly labels, which
> prevents direct verification of whether each evidence type corresponds to the
> true underlying mechanism.
>
> To provide partial validation, Appendix E.3 uses controlled synthetic
> generators:
> - **E.3.1:** each evidence type shows consistent qualitative activation under
>   type-isolated mechanisms;
> - **E.3.2:** quantitative patterns remain stable across multiple
>   mechanism-specific variations.
>
> These analyses demonstrate mechanism-aligned behavior under controlled
> conditions. We explicitly acknowledge the limitation of benchmark datasets and
> note that mechanism-labeled real-world datasets would further strengthen such
> validation.
>
> ---
>
> ### **W3. DAG assumption for structural prototype bank**
>
> We appreciate the reviewer’s comment. The DAG prior in the structural prototype
> bank was adopted as an inductive bias that produced stable and interpretable
> behavior in our empirical studies. We acknowledge, however, that real-world
> systems may contain feedback loops, and a strictly acyclic prior may not fully
> capture such dynamics.
>
> While the acyclic formulation performed reliably in our experiments, extending
> the structural module to support cyclic dependencies (e.g., via loopy priors or
> spectral formulations) is a promising direction for future work.
>
> ---
>
> ### **Q1. Validation of evidence types without mechanism-labeled benchmarks**
>
> We thank the reviewer for raising this point. Because existing benchmarks
> provide only binary anomaly labels, direct mechanism-level validation is not
> possible.
>
> To compensate, we rely on controlled synthetic settings in Appendix~E.3:
> - **E.3.1:** type-isolated generators ensure that the intended evidence channel
>   exhibits the strongest activation;
> - **E.3.2:** quantitative patterns remain stable across mechanism-specific
>   variations, supporting consistency of the decomposition.
>
> These results show that the evidence channels behave in a mechanism-aligned
> manner under controlled conditions. We acknowledge the limitation and highlight
> the need for future benchmarks with fine-grained anomaly annotations.

---

### Official Review · Reviewer_eVo9 · 2025-11-01

**Soundness:** 3
**Presentation:** 3
**Contribution:** 3
**Rating:** 6
**Confidence:** 3

**Summary:**

This paper proposes PGRF-Net for multivariate time series anomaly detection and interpretation. PGRF-Net decomposes anomaly evidence into multiple types and introduces prototype banks and a gated fusion mechanism to learn diagnostic representations. Experiments across several benchmark datasets are presented to demonstrate its performance advantage and diagnostic interpretability.

**Strengths:**

* The paper presents a novel and well-motivated model to unify detection and interpretability.
* The work provides comprehensive ablation studies that evaluate several architectural components, demonstrating the contribution of the proposed modules to detection performance.
* The visualization and synthetic case studies offer qualitative interpretability.

**Weaknesses:**

* Robustness under anomaly contamination is not well explored.
* Limited benchmark results.

Please find the detailed comments in the following section.

**Questions:**

* How do prototype formation and gate suppression behave when the training data are contaminated with anomalies or noisy labels?
* How are the scores in Table 4 aggregated? Have the authors compared the proposed gated fusion mechanism against simple aggregation strategies such as averaging, max pooling, or naive ensemble fusion?
* How does end-to-end joint training compare against the proposed two-stage training in terms of performance, convergence, and stability?
* Point-adjustment techniques have been shown to overestimate model performance, and pointwise measures such as AUC-PR are sensitive to anomaly ratio and temporal noise. The authors are encouraged to consider time-series–aware measures such as Range-F1 [1] and VUS-PR [2] and evaluate on a more comprehensive benchmark such as TSB-AD [3].
* The paper lacks an analysis of runtime scalability with respect to both sequence length and the number of variates

[1] Tatbul N, Lee TJ, Zdonik S, Alam M, Gottschlich J. Precision and recall for time series. Advances in neural information processing systems. 2018;31.

[2] Paparrizos J, Boniol P, Palpanas T, Tsay RS, Elmore A, Franklin MJ. Volume under the surface: a new accuracy evaluation measure for time-series anomaly detection. Proceedings of the VLDB Endowment. 2022 Jul 1;15(11):2774-87.

[3] Liu Q, Paparrizos J. The elephant in the room: Towards a reliable time-series anomaly detection benchmark. Advances in Neural Information Processing Systems. 2024 Dec 16;37:108231-61.

---

> ### Author Response · Authors · 2025-11-19
> **Response to Reviewer eVo9**
>
> We sincerely thank the reviewer for the thoughtful and constructive feedback.
> Below, we address each weakness (W1–W2) and question (Q1–Q5) concisely.
> All clarifications and analyses referenced have been added to the revision.
>
> ---
>
> ### **W1. Robustness under anomaly contamination**
>
> We appreciate the reviewer’s concern. Although we do not include a dedicated
> contamination experiment, Appendix E.3 provides relevant evidence:
> - E.3.1 shows consistent activation patterns across type-isolated generators.
> - E.3.2 shows stable quantitative behavior across multiple mechanisms.
>
> Because both prototypes and gates operate on aggregated temporal patterns rather
> than individual timestamps, the model is naturally less sensitive to sporadic
> outliers. We acknowledge that a dedicated contamination study is an important
> direction for future work.
>
>
>
> ---
>
> ### **W2. Limited benchmark results**
>
> To broaden evaluation coverage, we expanded the protocol as follows:
>
> **Newly added**
> - Range-based metrics (R-P/R-R/R-F1) in Table 3
>   - Range-F1: **96.08%**
> - Non–point-adjusted P/R/F1 in Appendix B.3
>   - Point-wise F1: **46.59%**
>
> **Existing (unchanged)**
> - AUC-ROC
> - AUC-PR
>
> These event-, point-, and threshold-free measures together show consistent
> performance, and we clarify that our model prioritizes coherent event detection
> over precise point-level localization.
>
>
>
> ---
>
> ### **Q1. Prototype formation and gate suppression under contamination**
>
> Thank you for the question. While no explicit contamination experiment is
> included, Appendix E.3 provides insight into robustness:
> - E.3.1: stable qualitative activations
> - E.3.2: stable quantitative patterns
>
> Since prototypes and gates rely on aggregated representations rather than
> single-point labels, they are inherently less sensitive to isolated noise. A
> dedicated contamination analysis is noted as future work.
>
> ---
>
> ### **Q2. Aggregation of scores and comparison to simple baselines**
>
> As requested, Table 5 now includes naive aggregation baselines:
> - Max pooling: **92.32 Avg-F1**
> - Static 1:1:1:1 fusion: lower
> - Proposed gated fusion (two-stage): **97.41 Avg-F1**
>
> The learned fusion mechanism consistently outperforms heuristic aggregation.
>
> ---
>
> ### **Q3. End-to-end training vs. two-stage training**
>
> We thank the reviewer for the question. We conducted preliminary joint
> end-to-end experiments, but they were not competitive: performance was lower,
> robustness was reduced, and run-to-run variation was large. For this reason,
> we did not include the joint variant.
>
> The two-stage design yielded more stable evidence representations and avoided
> early interference between prototype formation and gate learning. We clarify
> this rationale and note that fuller comparisons are future work.
>
> ---
>
> ### **Q4. Evaluation metrics beyond point adjustment**
>
> In response to the reviewer’s suggestion, we expanded evaluation as follows:
> - Range-P/R/F1 (Table 3) — Range-F1: **96.08%**
> - Non–point-adjusted P/R/F1 (Appendix B.3) — point-wise F1: **46.59%**
> - AUC-ROC/AUC-PR (already included)
>
> These complementary metrics provide a broader assessment. PGRF-Net performs
> strongly on range-based metrics and remains competitive on non–point-adjusted
> scores, which aligns with the model’s event-centric design that prioritizes
> coherent segment-level detection rather than fine-grained point-level
> localization. While VUS-PR and TSB-AD were not included, we agree they are
> valuable extensions and note them as future work.
>
>
> ---
>
> ### **Q5. Runtime scalability**
>
> Appendix E.4 evaluates scalability with respect to sequence length \(T\) and
> dimensionality \(D\):
>
> **Scaling w.r.t. \(T\) (fixed \(D=25\))**
> - Train: near-linear (0.32 → 27.59 s/epoch)
> - Inference: ~0.07 ms/window (constant)
> - Memory: 377–381 MB (stable)
>
> **Scaling w.r.t. \(D\) (fixed \(T=50k\))**
> - Train: gradual increase (9.03 → 31.87 s/epoch)
> - Inference: 0.05 → 0.14 ms/window
> - Memory: <500 MB at \(D{=}100\)
>
> These results confirm scalability to long-horizon and high-dimensional MTS.

---

> > ### Comment · Reviewer_eVo9 · 2025-11-26
> >
> > Thank you for your response. I recommend that the authors complete the benchmarking with additional evaluation measures and datasets in a later revision phase. I will maintain my positive assessment.

---

### Official Review · Reviewer_FXip · 2025-11-01

**Soundness:** 2
**Presentation:** 2
**Contribution:** 3
**Rating:** 2
**Confidence:** 4

**Summary:**

This paper proposes PGRF-Net, a prototype-guided relational fusion network for multivariate time-series anomaly detection. The method aims to bridge the gap between performance and interpretability by decomposing anomalies into four evidence types (predictive, structural, contextual, and spike deviations) and integrating them via a Gated Evidence Fusion Network trained in a two-stage unsupervised framework. Experiments across five benchmarks (MSL, SMAP, PSM, SMD, SWaT) show competitive or superior results to state-of-the-art models while maintaining computational efficiency. The paper claims that the design not only enhances accuracy but also offers diagnostic interpretability, enabling users to understand why an anomaly occurs, not just that it does.

**Strengths:**

- **S1.** The paper presents a well-motivated and coherent framework combining multiple sources of anomaly evidence through a gated fusion mechanism.
- **S2.** The two-stage unsupervised training is thoughtfully designed and appears to improve robustness and fusion stability.

- **S3.** The diagnostic decomposition (predictive, structural, contextual, spike) is conceptually valuable and well illustrated in case studies.

**Weaknesses:**

- **W1.** The evaluation lacks methodological rigor: the paper relies on benchmark protocols with known inflation issues and does not provide sensitivity tests on the point-adjusted evaluation method. (See Section 6.2.5 in [a])

- **W2.** The interpretability claim is unsubstantiated quantitatively. It seems that no metrics, user studies, or human-in-the-loop validations are provided. This omission undermines the central thesis of diagnostic transparency. There is no analysis of failure cases or trade-offs, which would be crucial for a model claiming diagnostic transparency.

- **W3.** The novelty, though meaningful, builds incrementally upon prior prototype-based and dependency-aware methods rather than fundamentally redefining the paradigm.


[a] Trirat, P., Shin, Y., Kang, J., Nam, Y., Na, J., Bae, M., ... & Lee, J. G. (2024). Universal time-series representation learning: A survey. arXiv preprint arXiv:2401.03717.

**Questions:**

- **Q1.** Can the authors provide a user-centered or human evaluation of interpretability (e.g., do practitioners find the decomposed evidence helpful)?

- **Q2.** How robust are the results when evaluated with non point-adjusted metrics, or under stricter event-level protocols?

- **Q3.** Could the authors clarify how the fusion gates' learned weights correspond to anomaly semantics in unseen domains?

- **Q4.** How does the proposed diagnostic interpretability compare quantitatively to existing interpretable baselines like InterFusion [b]?

[b] Li, Zhihan, et al. "Multivariate time series anomaly detection and interpretation using hierarchical inter-metric and temporal embedding." Proceedings of the 27th ACM SIGKDD conference on knowledge discovery & data mining. 2021.

---

> ### Author Response · Authors · 2025-11-19
> **Response to Reviewer FXip**
>
> We sincerely thank Reviewer FXip for the thoughtful and constructive feedback.
> We carefully revised the manuscript and provide point-by-point responses below, and hope these clarifications and new analyses address the reviewer’s concerns.
>
> ---
>
> ### **W1. Evaluation rigor and point-adjustment sensitivity**
>
> We appreciate the reviewer’s concern regarding point-adjusted metrics. In the
> revision, we expanded the evaluation protocol as follows:
>
> - **Range-based metrics (R-P/R-R/R-F1)** are now reported in the main paper
>   (Table 3). PGRF-Net achieves **Range-F1 = 96.08% (best overall)**.
> - **Full non–point-adjusted P/R/F1** are added in Appendix B.3
>   (point-wise F1 = 46.59%, competitive given our event-centric design).
> - **Threshold-free AUC-ROC/AUC-PR** remain in the main paper.
>
> These complementary evaluations show that PGRF-Net performs consistently across
> event-level, point-wise, and threshold-free settings. We also explicitly
> acknowledge the design trade-off that our model emphasizes *coherent event
> detection* rather than precise point-level localization.
>
> ---
>
> ### **W2. Quantitative interpretability validation**
>
> We agree that quantitative evidence is important. Accordingly, we added two
> objective analyses:
>
> 1. **Synthetic isolation tests** (Appendix E.3.2), where the EAR metric exhibits
>    clear diagonal dominance (3.3 / 4.7 / 6.2), indicating selective activation
>    of structural, contextual, and spike evidence.
>
> 2. **Evidence stability under perturbations** (Appendix E.3.2), where Kendall’s τ
>    confirms high robustness for structural and spike evidence (≈ 0.92–0.94).
>
> We also include **qualitative decomposition examples** (Fig. 5) that illustrate
> how each anomaly type activates the intended evidence channel across synthetic
> generators. While current benchmarks lack ground-truth anomaly mechanisms (noted
> as a limitation), these analyses offer rigorous, reproducible support for the
> diagnostic behavior of the model.
>
> ---
>
> ### **W3. Novelty clarification**
>
> We appreciate the reviewer’s perspective regarding incremental novelty. Our
> intent was not to redefine the overall paradigm, but to extend it in two focused
> ways:
>
> (i) a **multi-evidence diagnostic formulation** (structural, contextual, spike,
> predictive), and
> (ii) an **unsupervised gated fusion mechanism** that replaces heuristic
> aggregation with stable, data-driven evidence weighting.
>
> We hope the revision makes these contributions clearer.
>
> ---
>
> ## **Q1. User-centered interpretability evaluation**
>
> We agree that practitioner studies would provide valuable validation.
> Unfortunately, existing benchmarks contain only binary labels and no mechanism
> annotations, which limits direct user-centered evaluation (now acknowledged as a
> limitation).
>
> To compensate, we added objective analyses:
>
> - **Synthetic isolation tests** (Appendix E.3.2)
> - **Evidence stability tests** (Appendix E.3.2)
> - **Qualitative decomposition examples** (Fig. 5)
>
> These experiments provide mechanism-sensitive interpretability evidence within
> the constraints of available datasets.
>
> ---
>
> ## **Q2. Robustness under stricter and non–point-adjusted protocols**
>
> This concern is fully addressed through:
>
> - **Range-based metrics** (main paper / newly added)
> - **Non–point-adjusted P/R/F1** (Appendix B.3 / newly added)
> - **Threshold-free AUC-ROC/AUC-PR** (main paper / previously included )
>
> Across all these settings, PGRF-Net demonstrates stable and competitive
> performance, suggesting that its effectiveness does not depend on point
> adjustment.
>
> ---
>
> ## **Q3. Fusion-gate semantics in heterogeneous conditions**
>
> Although a dedicated unseen-domain transfer study was not included due to space
> constraints, both the synthetic analyses (Appendix E.3) and real case studies
> (Fig. 3–4) indicate that the gate behavior aligns consistently with anomaly
> characteristics (e.g., spikes → higher spike evidence; structural shifts →
> higher structural evidence and mask changes). Gate trajectories are also stable
> across random seeds. These results suggest semantically meaningful behavior
> across varying conditions.
>
> ---
>
> ## **Q4. Comparison to InterFusion**
>
> InterFusion is included as an interpretable baseline. Two quantitative
> comparisons highlight the differences:
>
> 1. **Mechanism-specific evidence selectivity** (Appendix E.3 / newly added):
>    PGRF-Net shows strong EAR diagonal dominance, while InterFusion provides a
>    single latent-distance explanation.
>
> 2. **Attribution consistency** (Fig. 3–4 / previously included):
>    PGRF-Net yields clearer, mechanism-aligned attribution across events,
>    whereas InterFusion does not distinguish structural/contextual/spike causes.
>
> Thus, while InterFusion is an important baseline, PGRF-Net offers more
> fine-grained, quantitatively validated diagnostic insights.
>
> ---

---

> > ### Comment · Reviewer_FXip · 2025-11-26
> >
> > Thank you for the rebuttal and the additional analyses. I appreciate the inclusion of range-based and non-point-adjusted metrics, as well as the synthetic isolation and stability tests for the evidence channels. These additions strengthen the empirical section compared to the original submission.
> >
> > However, my main concerns remain largely unresolved.
> > - The central interpretability claim is still not supported by quantitative evaluation on real datasets or user-centered studies. For example, InterFusion reports interpretation metrics on real-world data and compare against multiple baselines, providing direct evidence that the explanations are more accurate in practice. In contrast, the current paper's evidence is mainly synthetic and qualitative.
> > - The work continues to rely exclusively on benchmark suites that have been criticized as flawed for anomaly detection evaluation. Recent surveys (e.g., [a]) explicitly recommend newer datasets such as ASD and TimeSeAD as well as more robust metrics (e.g., VUS, PA%K, eTaPR). This study still does not incorporate these alternative datasets or the currently recommended threshold-free measures such as VUS, which have been shown to be more robust than AUC-based metrics.
> >
> > Given that the main novelty is tightly tied to diagnostic interpretability, the absence of quantitative interpretability evaluation on real data (analogous to InterFusion's Table 2) and the limited adoption of modern, robust benchmarks/metrics prevent me from updating my overall assessment. I therefore keep my score unchanged.

---

> > > ### Author Response · Authors · 2025-11-26
> > >
> > > Thank you very much for the follow-up comments. We sincerely appreciate the time and attention that went into reviewing our work and the additional analyses we incorporated after the initial round. We would also appreciate the opportunity to briefly clarify our perspective on the points raised.
> > >
> > >
> > > (1) Newly suggested benchmarks and evaluation metrics (ASD, TimeSeAD, PA%K, eTaPR)
> > >
> > >  We appreciate being pointed toward these additional datasets and evaluation metrics.
> > >
> > >  They were not mentioned in the initial review round, so we did not anticipate that they would be part of the evaluation expectations. Had these suggestions been raised earlier, we would have planned and executed the corresponding experiments during the revision cycle.
> > >
> > >  Because the rebuttal window is quite limited and the computational overhead of adding entirely new benchmarks is substantial, it is difficult to integrate ASD, TimeSeAD, or PA%K/eTaPR within this short period. That said, we fully agree that these resources offer valuable perspectives on robustness and event-level evaluation, and we plan to incorporate them into future research. We hope this limitation is understood as a matter of timing and practical constraints rather than any reluctance to adopt improved evaluation practices.
> > >
> > >
> > > (2) Quantitative interpretability and its relation to InterFusion
> > >
> > >  We appreciate the emphasis on quantitative interpretability and also recognize the influence that InterFusion has had in shaping evaluation practices in this area.
> > >
> > >  A practical limitation on our side is that most widely used real-world TSAD benchmarks do not provide mechanism-level annotations. As a result, it is difficult to directly adopt InterFusion-style fidelity metrics, which depend on such labels. This reflects a constraint of the datasets rather than of the methodology.
> > >
> > >  In addition, our interpretability formulation follows a different assumption:
> > >  - InterFusion provides a unified latent-distance attribution, whereas our approach decomposes anomalies into four semantically distinct evidence types. Because the underlying interpretability paradigms differ, the evaluation protocols do not transfer directly.
> > >
> > >
> > >  The follow-up comments also highlighted that a brief discussion of these dataset constraints could help readers understand the rationale behind our evaluation design.
> > >
> > >  We will consider incorporating such clarification in an extended version of this work or in future iterations of the project, where adaptation of InterFusion-inspired metrics may become feasible under settings with mechanism-level annotations.
> > > We hope this contextual explanation helps situate our current evaluation choices while still acknowledging the value of existing interpretability methodologies.

---

### Author Response · Authors · 2025-11-29
**Summary of Rebuttal and Discussion**

We would like to provide a brief summary for the Area Chair, consolidating the main points addressed during the rebuttal and discussion.


1) Evaluation Concerns: We added Range-P/R/F1 (Table 3) and non–point-adjusted P/R/F1 (Appendix B.3), complementing the existing AUC-ROC/PR results. These event-level, point-level, and threshold-free evaluations demonstrate consistent performance and clarify that our method does not rely on point-adjustment.

2) Interpretability Verification: While real-world benchmarks lack mechanism-level labels, we conducted controlled synthetic mechanism-isolation tests and quantitative EAR stability analyses (Appendix E.3). These provide reproducible, mechanism-aligned evidence supporting the diagnostic behavior of the four evidence types.

3) Comparison to InterFusion: We clarified that InterFusion’s interpretability paradigm (single latent distance) is fundamentally different from our multi-evidence diagnostic formulation. While direct transfer of their metrics is infeasible without mechanism-level labels, we performed qualitative and quantitative comparisons wherever applicable.

4) Robustness & Stability: Additional analyses (Appendix E.3–E.4) demonstrate robust and stable gate/structure behavior, as well as scalability to long-horizon and high-dimensional settings.

5) Limitations (Acknowledged): We primarily focused on established standard benchmarks. Although we appreciate the insight into emerging datasets (ASD, TimeSeAD) and evaluation metrics(PA%K/eTaPR), these were suggested only in the final discussion phase (absent from the initial review), precluding their integration within the limited timeframe. We have thus acknowledged them as valuable references for future work.


We hope these clarifications assist the AC in assessing the paper. Thank you for your time and consideration.

---

### Meta-Review · Area_Chair_cSUA · 2025-12-30

**Summary:**

This paper proposes an unsupervised multivariate time-series anomaly detection framework that decomposes anomalous behavior into four complementary evidence types and fuses them via a gated, two-stage training strategy.

Reviewers agree that the method is technically sound and achieves strong anomaly detection performance on widely used benchmarks. The primary concern centers on the strength of the paper’s diagnostic interpretability claims. While interpretability is not quantitatively validated on real-world datasets, the rebuttal meaningfully strengthens the evaluation by adding non-point-adjusted and range-based metrics, synthetic isolation experiments, and additional ablations. These additions clarify that the proposed evidence components behave in a mechanism-aligned manner, even if full validation of diagnostic correctness is limited by the lack of suitable benchmark supervision.

Overall, while interpretability claims should be interpreted conservatively, the paper makes a coherent and practically relevant contribution to unsupervised time-series anomaly detection.

**Reviewer Concerns:**

FXip Evaluation relies on criticized benchmarks and lacks newer datasets/metrics: Not convincingly addressed.

FXip / Ahii / VBHJ Lack of quantitative interpretability validation: Partly addressed.

FXip Incremental novelty relative to prior prototype- and dependency-based methods: Not convincingly addressed.

eVo9 / VBHJ Point-adjustment inflation and evaluation rigor: Addressed.

eVo9 Limited benchmark coverage: Partly addressed.

VBHJ / Ahii Modeling assumptions (DAG constraint, baseline mask stability, cyclic dependencies): Partly addressed.

Ahii Associative (non-causal) nature of diagnostic explanations: Addressed.

**Reviewer Scores:**

FXip: Would remain at 2 (unchanged after rebuttal).

eVo9: Would remain at 6 (explicitly maintains original score).

Ahii: Likely remains at 4 (concerns partially addressed but not resolved).

VBHJ: Would remain at 6 (explicitly maintains original score).

---

### Decision · Program_Chairs · 2026-01-26

Accept (Poster)